# PRL: PROMPTS FROM REINFORCEMENT LEARNING

## ABSTRACT

Effective prompt engineering remains a central challenge in fully harnessing the capabilities of LLMs. While well-designed prompts can dramatically enhance performance, crafting them typically demands expert intuition and a nuanced understanding of the task. Moreover, the most impactful prompts often hinge on subtle semantic cues, ones that may elude human perception but are crucial for guiding LLM behavior. In this paper, we introduce PRL (Prompts from Reinforcement Learning), a novel RL-based approach for automatic prompt generation. Unlike previous methods, PRL can produce novel few-shot examples that were not seen during training. Our approach achieves state-of-the-art performance across a range of benchmarks, including text classification, simplification, summarization, and GSM8K. On the classification task, it surpasses prior methods by 2.58% over APE and 1.00% over EvoPrompt. Additionally, it improves the average ROUGE scores on the summarization task by 4.32 over APE and by 2.12 over EvoPrompt and the SARI score on simplification by 6.93 over APE and by 6.01 over EvoPrompt. On the GSM8K mathematical reasoning benchmark, PRL further improves accuracy by 2.72% over APE and by 4.53% over EvoPrompt. We will make our implementation publicly available upon acceptance.

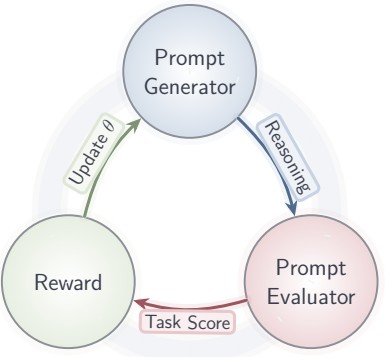

| Method | Gen. | Ref. | Few-shot |
|---|---|---|---|
| Manual Instr. | ✗ | ✗ | ✗ |
| APE | ✓ | ✗ | ✗ |
| EvoPrompt | ✓ | ✓ | ✗ |
| APO | ✓ | ✓ | ∼ |
| PRL | ✓ | ✓ | ✓ |

✓ supported ✗ not supported ∼ limited

Figure 1: Left: Our RL-based prompt optimization cycle (overview). Right: Comparison of prompt-engineering methods. PRL automates both prompt generation and refinement and, uniquely, synthesizes novel task-specific few-shot examples. The yellow tilde (∼) for APO indicates *limited* few-shot support—its examples are drawn from training data, which restricts performance—whereas PRL creates new instances not seen during training.

## 1 INTRODUCTION

Prompt engineering has emerged as a key technique for enhancing the performance of LLMs (Sahoo et al., 2024; Chen et al., 2023). By crafting precise input prompts, LLMs can be guided to perform complex tasks without requiring additional fine-tuning. However, the effectiveness of a prompt often hinges on subtle phrasing. As shown by Razavi et al. (2025), even minor rewordings can significantly alter model predictions, underscoring the fragility of prompt-based control. Moreover, the DeepSeek-R1 paper Guo et al. (2025) states that even a model as large as DeepSeek-R1, with 671 billion parameters, is sensitive to prompts.

Few-shot prompting, in which a prompt includes a small set of input-output examples, is another widely used approach to guide LLMs. While often beneficial, Reynolds & McDonell (2021) demonstrate that zero-shot prompting can sometimes outperform few-shot approaches, suggesting that the usefulness of examples may depend on task familiarity or pretraining exposure. These findings collectively highlight the challenge of designing effective prompts and motivate the need for automated, task-specific prompt optimization.

Recent work has explored automatic prompt generation (Zhang et al., 2022) and refinement (Guo et al., 2023; Pryzant et al., 2023). Existing methods, with the partial exception of (Pryzant et al., 2023), fail to integrate tailored few-shot examples. We propose PRL, a RL-based prompt optimization algorithm based on reinforcement learning. PRL is capable of automatically determining whether few-shot examples should be included and if so, to create them to maximize task performance. Interestingly, the incorporation of few-shot examples is spontaneously emerging during the prompt generation training and is not explicitly encouraged. Additionally, PRL incorporates a reasoning phase prior to prompt generation, where the model first produces a rationale to guide its final output. We additionally mitigate training instability and noisy feedback with a prompt selection strategy that improves robustness in the limited data setting.

**Contributions.** This paper makes the following contributions:

- We propose PRL, to our knowledge the first RL-based prompt optimization method capable of generating and selecting novel task-specific few-shot examples.
- We demonstrate the effectiveness of PRL across text classification, summarization, simplification and mathematical reasoning tasks.
- We show that integrating explicit reasoning before answer generation significantly boosts performance, echoing findings by Guo et al. (2025).
- We provide detailed ablation studies to evaluate the impact of each component.
- Our results suggest that RL-based optimization naturally leads to the emergence of few-shot prompting behavior.

## 2 RELATED WORK

**Prompt Engineering** enhances model performance without retraining, offering a cost-effective solution. Chain-of-Thought (CoT) prompting (Wei et al., 2022) improves reasoning by including intermediate steps. Tree-of-Thought (ToT) (Yao et al., 2023) extends this by exploring multiple reasoning paths, while Program-of-Thoughts (Chen et al., 2022) and Graph-of-Thoughts (Besta et al., 2024) further enrich prompts using programmatic and graph structures.

Few-shot prompting (Brown et al., 2020) improves performance by embedding task examples in prompts, proving effective in areas like puzzle solving and evidence extraction (Xu et al., 2023; Greenblatt, 2024; Sivarajkumar et al., 2024). However, such examples can sometimes hurt performance (Reynolds & McDonell, 2021), making their use highly task-dependent. Our method automatically learns whether and how to include few-shot examples based on task performance.

**Automated Prompt Engineering** improves task performance by replacing manual prompt design with automated methods. The work most closely related to ours is RLPrompt Deng et al. (2022), which also uses RL to automatically generate prompts. However, the authors only learn a small policy network and are restricted to short prompts of at most five tokens. Additionally, these prompts are ungrammatical gibberish text, hence lack interpretability. Moreover, their pipeline is more complex and involves reward stabilization. The Automatic Prompt Engineer (APE) Zhou et al. (2022) generates prompt candidates from input-output examples and filters them based on performance. As no gains were observed from in-sample refinement, APE remains a pure generation method. Pryzant et al. (2023) introduced Automatic Prompt Optimization (APO), which iteratively improves prompts using natural language critiques, simulating gradient descent. APO includes few-shot examples in its prompt, but is restricted to examples seen during training. It enhances efficiency via minibatching, beam search, and bandit selection. Guo et al. (2023) proposed EvoPrompt, which evolves a population of prompts with LLMs and evolutionary operators, achieving strong results without needing model gradients. PRL is, to our knowledge, the only method that can create novel few-shot examples not seen during training.

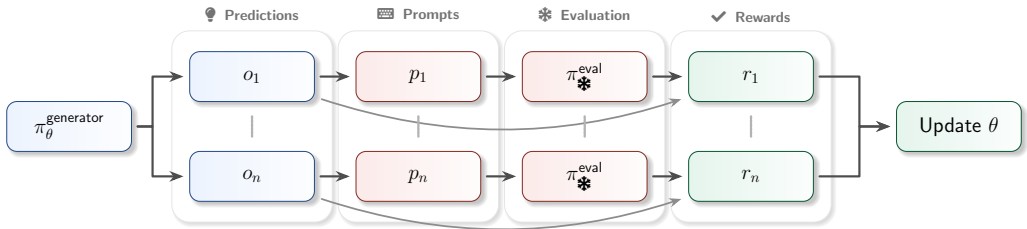

Figure 2: Training scheme of PRL. First, the Prompt Generator $\pi_\theta^{\text{generator}}$ generates a set of outputs $o_1, \ldots, o_n$ (reasoning + generated prompt) from which the corresponding prompts $p_1, \ldots, p_n$ are extracted. Each prompt is then evaluated by the Evaluation Model $\pi^{\text{eval}}$ (a language model with frozen parameters), which produces corresponding answers. These answers, along with the outputs from the Prompt Generator, are used to compute rewards $r_1, \ldots, r_n$. Finally, the rewards are used to update the parameters of the Prompt Generator through RL.

## 3 METHOD

Our method comprises the following components:

- **Prompt Generator:** A trainable language model that generates prompts with help of a reasoning process, see the prompt formats in Appendix B.
- **Evaluation Model:** A frozen LLM that produces an answer based on the generated prompt.
- **Prompt Selection:** We learn the prompt generator through RL with a reward incorporating both formatting and task performance. We choose the best overall prompt by regularly querying the prompt generator model for prompt candidates and evaluate those.

We now provide a detailed description of each component of our method.

**Reward Function** Our reward signal is composed of two parts: formatting rewards for the Prompt Generator and task performance rewards for the evaluation model.

Prompt Generator Reward: We assess the format of the Prompt Generator's output using the following scheme:

- A reward of $\frac{r_{\text{token}}}{4}$ is assigned for the correct usage of each of the four key tokens: `<think>`, `</think>`, `<answer>`, and `</answer>`, provided that each token appears exactly once. If all tokens are used correctly, the model receives the full token reward of $r_{\text{token}}$. This encourages the model to output both a reasoning trace and a final answer, each clearly marked and occurring only once.
- An additional reward of $r_{\text{structure}}$ is granted if the generated response exactly matches the required structure: `<think>` reasoning trace `</think>` `<answer>` final answer `</answer>`. This structural reward ensures that the model produces a well-formed response consisting of a reasoning trace followed by a final answer and nothing beyond this expected format.

Evaluation Model Reward: To assess the utility of the prompts, we assign:

- A reward of $r_{\text{format}}$ is assigned if the Evaluation Model's response follows the required format. This reward is applicable only when the output must adhere to a specific structure, such as selecting a predefined class; otherwise, $r_{\text{format}}$ is set to 0.
- A reward of $r_{\text{alignment}}$ is granted if the model's response is factually correct or aligned with the intended task objective. This reward is typically based on task-specific metrics such as accuracy or any other evaluation criterion that reflects successful performance.

These rewards are task-specific and will be discussed in detail in the experimental section for each respective task.

The overall reward function $R$ is the sum of all elementary applicable rewards $r_{\text{token}}, r_{\text{structure}}, r_{\text{format}}$ and $r_{\text{alignment}}$.

**Prompt Generator** is a language model designed to refine a given base prompt. First, it generates a reasoning trace about the refinement process, followed by the production of the final prompt (an example is shown in Appendix B). During training, the Prompt Generator learns task semantics and produces suitable prompts, potentially incorporating examples for few-shot prompting.

At each training step, the Prompt Generator produces a set of outputs $o_1, \ldots, o_n$ from which candidate prompts $p_1, \ldots, p_n$ are extracted. These prompts are then passed to the Evaluation Model, which generates answers conditioned on each prompt. We denote the Prompt Generator as $\pi_\theta^{\text{generator}}$.

**Evaluation Model.** The *Evaluation Model* assesses the quality of each prompt generated by the Prompt Generator by evaluating its performance on a randomly sampled subset of the training data. For each prompt, a reward is computed for every observation based on its effectiveness, and these rewards are averaged to obtain a final score for the prompt. The Evaluation Model is implemented as a frozen language model, used exclusively for inference. We denote the Evaluation Model as $\pi^{\text{eval}}$.

**Remark 1.** *We have chosen to freeze the evaluation model, since then our method can in principle be used with closed-source LLMs. It also conforms to the setting of existing work that did not finetune the LLM executing the prompt.*

**Optimization** After obtaining a reward for each prompt, we optimize the Prompt Generator using the Group Relative Policy Optimization (GRPO) update rule (Shao et al., 2024). A key advantage of GRPO is that it eliminates the need for a separate critic network, significantly reducing memory consumption during the reinforcement learning process. The illustration of this process can be found in Figure 2.

**Prompt Selection** As the prompt generator evolves during training and each version generates multiple prompts, we obtain a large selection of candidate prompts. We sample in a regular interval a number of prompts and test them on the validation set and keep the overall best one according to the task metric used. The full algorithm is showcased in the Appendix A.

## 4 EXPERIMENTS

**Experimental Setup** We test the performance of PRL on four task types: **classification**, **summarization**, **simplification**, and **GSM8K mathematical reasoning**. As Prompt Generator and Evaluation Model we choose Qwen2.5-7B-Instruct (Yang et al. (2024)). Each model is trained separately for each task and dataset. Unless otherwise specified, all experiments are conducted over 48 hours using two NVIDIA A100 GPUs (40 GB each). We fine-tune our models using GRPO (Zhao et al. (2024)) with parameters $\epsilon = 0.2$, $\beta = 0.04$ and weight decay equal to 0.1. We also use Low-Rank Adaptation (LoRA) (Hu et al. (2022)) with a learning rate of $1 \times 10^{-6}$, setting $\alpha = 32$ and rank $r = 8$. During training, we sample $n = 4$ prompts per iteration and perform Prompt Selection every 100 iterations.

To ensure fair comparison during Prompt Selection, we adopt the same scoring function as EvoPrompt and use an identical validation dataset during the prompt selection process. Across all tasks we use reward parameters $r_{\text{token}} = r_{\text{structure}} = 0.75$, unless otherwise stated. The rewards $r_{\text{format}}$ and $r_{\text{alignment}}$ are task-specific and will be defined below.

**Baseline Methods** We benchmark PRL against both human-written task-specific prompts and a range of general-purpose prompt engineering algorithms.

- `MI (Manual Instruction)` (Zhang et al., 2022): Manually crafted instructions to fine-tune large language models, aiming to enhance their performance on specific tasks through human-written prompts.
- `NI (Natural Instruction)` (Mishra et al., 2021): NI comprises a diverse set of 61 NLP tasks, each accompanied by human-authored instructions. It is designed to evaluate models' abilities to generalize across tasks by understanding and following natural language instructions.
- `APE (Automatic Prompt Engineer)` (Zhou et al., 2022): APE introduces a framework for automatically generating and selecting prompts. It leverages large language models to create candidate instructions and selects the most effective ones based on performance evaluations.
- `APO (Automatic Prompt Optimization)` (Pryzant et al., 2023): APO presents a method for optimizing prompts by iteratively refining them using feedback mechanisms. It

treats prompt optimization as a gradient-free problem, employing techniques like beam search to enhance prompt effectiveness.

- `EvoPrompt` (Guo et al., 2023): EvoPrompt applies evolutionary algorithms to optimize discrete prompts for large language models. It evolves a population of prompts through selection, mutation, and crossover operations to discover high-performing prompts without requiring gradient information.
  - `DE (Differential Evolution)`: This variant employs differential evolution strategies to explore the prompt space.
  - `GA (Genetic Algorithm)`: This approach utilizes genetic algorithms to evolve prompts by simulating natural selection processes, including selection, crossover and mutation to optimize prompt quality over successive generations.

We present a comparison of various methods in Figure 1 (right). To ensure a fair comparison, we utilize the Qwen2.5-7B-Instruct model across all methods, serving both as the prompt generator and the Evaluation Model. Due to the crucial differences between PRL and RLPrompt (Deng et al., 2022), which made the latter impossible to reproduce fairly, we did not include RLPrompt in this section. However, we compared the prompts produced by the two methods. Details on reproduction and the comparison with RLPrompt are provided in Appendix E.

**Classification**    For this task, we evaluate our method on a variety of datasets, including:

- **Binary sentiment classification**: SST-2 (Socher et al. (2013)), MR (Pang & Lee (2005)), CR (Hu & Liu (2004)). These datasets involve classifying whether the semantic meaning of a sentence is `positive` or `negative`.
- **Multiclass sentiment classification**: SST-5 (Socher et al. (2013)) requires classifying a sentence into one of five sentiment categories: `terrible`, `bad`, `okay`, `good`, or `great`. This is more challenging than binary sentiment classification as it involves recognizing more nuanced emotional intensities.
- **Question classification**: TREC Voorhees & Tice (2000)) requires to categorize a question into one of six classes: `Description`, `Entity`, `Expression`, `Human`, `Location`, or `Number`. This task evaluates the model's understanding of the semantic type of the question.
- **News classification**: AG's News (Zhang et al. (2015)) requires to clasify news articles into one of four topics: `World`, `Sports`, `Business`, or `Tech`.
- **Subjectivity classification**: SUBJ (Pang & Lee (2004)) asks to determine whether a sentence is `subjective` or `objective`.

Table 1: Accuracy on classification tasks, averaged over three runs. Colours mark the best (red), second-best (orange) and third-best (yellow) numbers in each column; minor differences ($\leq 0.05$) are treated as ties. The right-most column shows the mean accuracy of each method across the seven datasets.

| Method / Dataset | SST-2 | CR | MR | SST-5 | AG's News | TREC | Subj | Avg |
|---|---|---|---|---|---|---|---|---|
| MI | 92.70 | 87.25 | 87.40 | 52.31 | 82.29 | 69.20 | 57.95 | 75.59 |
| NI | 95.77 | 91.50 | 90.85 | 51.90 | 83.43 | 66.60 | 68.10 | 78.31 |
| APO | $93.71_{\pm 0.25}$ | $93.48_{\pm 0.24}$ | $89.97_{\pm 1.37}$ | $53.94_{\pm 0.29}$ | $83.73_{\pm 0.31}$ | $71.30_{\pm 1.90}$ | $69.80_{\pm 5.96}$ | 79.42 |
| APE | $91.23_{\pm 0.66}$ | $92.87_{\pm 0.02}$ | $89.90_{\pm 0.94}$ | $49.37_{\pm 5.66}$ | $82.58_{\pm 1.20}$ | $77.07_{\pm 1.61}$ | $73.92_{\pm 1.39}$ | 79.56 |
| GA | $94.65_{\pm 1.04}$ | $92.75_{\pm 0.40}$ | $90.45_{\pm 0.72}$ | $53.76_{\pm 1.13}$ | $82.24_{\pm 1.00}$ | $79.20_{\pm 2.83}$ | $74.93_{\pm 3.12}$ | 81.14 |
| DE | $93.29_{\pm 0.34}$ | $93.38_{\pm 0.19}$ | $89.98_{\pm 0.24}$ | $55.25_{\pm 0.37}$ | $82.18_{\pm 1.04}$ | $76.47_{\pm 0.38}$ | $73.08_{\pm 4.95}$ | 80.52 |
| PRL (–PS) (ours) | $95.98_{\pm 0.19}$ | $92.17_{\pm 0.02}$ | $90.72_{\pm 0.05}$ | $54.80_{\pm 1.10}$ | $83.84_{\pm 0.33}$ | $72.00_{\pm 0.86}$ | $66.98_{\pm 2.86}$ | 79.50 |
| PRL (ours) | $96.32_{\pm 0.04}$ | $92.83_{\pm 0.24}$ | $91.27_{\pm 0.05}$ | $56.21_{\pm 0.15}$ | $84.36_{\pm 0.08}$ | $77.07_{\pm 2.36}$ | $76.90_{\pm 0.95}$ | 82.14 |

We apply a unified reward function across all classification tasks, with reward parameters set as $r_{\text{format}} = r_{\text{alignment}} = 1$. The component $r_{\text{format}}$ is specifically awarded when the Evaluation Model's output is a valid label, i.e. one that belongs to the task's set of permissible labels.

For example, in binary sentiment classification, a reward of $+1$ is given if the output is either `positive` or `negative`. This encourages the Prompt Generator to produce prompts that guide the Evaluation Model toward correct, task-appropriate responses.

The scoring function $f$ used in all classification tasks is accuracy. We set the number of test prompts to $n_{\text{test}} = 10$. For most tasks we sample a subset of 100 samples of our training set. For CR and AG's

News, due to longer average sentence lengths (which increases training and evaluation time), we reduce this to 30 samples.

We present our results in Table 1, where our method achieves state-of-the-art performance on all classification datasets. Notably, on the subjectivity classification task, our approach improves accuracy by 19% compared to the manual prompt baseline.

Figure 3 presents a comparison of the manual prompt, the PRL-generated prompt, and the EvoPrompt-generated prompt for the SUBJ classification task. As shown, the prompt generated by PRL is more detailed and explicit, providing clearer guidance for the model. Moreover, it is automatically tailored with task-specific few-shot examples, which contributes to its superior performance.

The remaining prompts for other classification tasks are included in Appendix C. As illustrated, all of these prompts incorporate few-shot examples, emphasizing the critical role of few-shot prompting in text classification. Interestingly, the examples generated by PRL do not appear in the training set, indicating that the model is able to synthesize relevant and task-aligned examples autonomously. This is in contrast to APO, which also can incorporate few-shot examples, but which are always selected from the training set. Specifically, few-shot examples in APO are selected from training samples which were incorrectly classified.

**Remark 2.** *In EvoPrompt (Guo et al., 2023) the task accuracy is computed by extracting the corresponding word for the classification from the full response. Hence, even when additional text is generated and hence the output does not strictly conform to the desired format, often a classification can still be obtained. In our work we train and evaluate all baselines by only accepting a response that is comprised of a single word denoting the classification. For summarization and simplification tasks we do not modify the EvoPrompt training and evaluation process.*

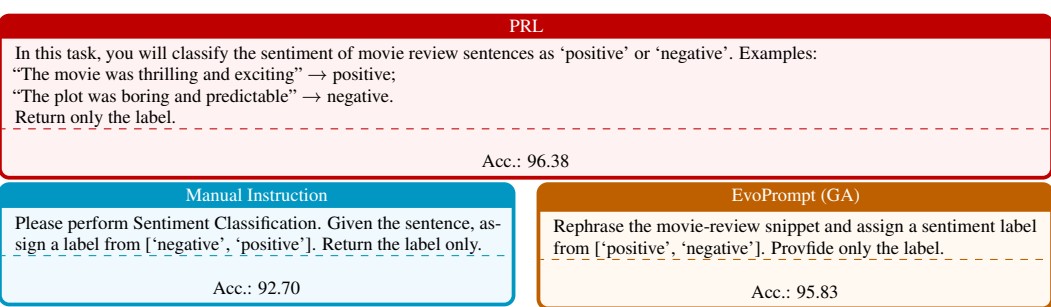

Figure 3: Comparison of a manual instruction, the best PRL prompt, and the best EvoPrompt prompt along with their accuracies on SST-2 task.

**Summarization**  We evaluate PRL on a summarization task, where the model is required to extract and condense the most important information from a given text. The objective is to generate a concise summary that preserves key content while omitting irrelevant or redundant details.

Our experiments are conducted on the SAMSUM dataset (Gliwa et al., 2019), which comprises English-language chat dialogues resembling real-life messenger conversations. These dialogues were synthetically generated by linguists to reflect informal, everyday exchanges and are accompanied by manually written abstractive summaries. To assess summarization performance, we adopt the widely used ROUGE metrics (Lin (2004)), reporting scores for the following variants:

Table 2: Text summarization results.

| Method | ROUGE-1 | ROUGE-2 | ROUGE-L |
|--------|---------|---------|---------|
| MI | 32.76 | 10.39 | 28.97 |
| APE | $37.12_{\pm 2.02}$ | $12.97_{\pm 0.74}$ | $33.32_{\pm 1.68}$ |
| GA | $39.69_{\pm 1.76}$ | $14.47_{\pm 1.00}$ | $35.84_{\pm 1.63}$ |
| DE | $33.91_{\pm 4.04}$ | $12.53_{\pm 1.47}$ | $31.05_{\pm 3.79}$ |
| PRL | $42.47_{\pm 0.83}$ | $16.17_{\pm 0.24}$ | $37.73_{\pm 0.36}$ |

- **ROUGE-1**: Measures the overlap of individual words (unigrams) between the generated summary and the reference summary, focusing on content selection.
- **ROUGE-2**: Measures the overlap of consecutive word pairs (bigrams), capturing the ability of the model to preserve local coherence and phrasing.

- **ROUGE-L**: Measures the longest common subsequence of words between the generated and reference summaries, evaluating the overall fluency and structure alignment.

For this task, we set $r_{\text{format}} = 0$, as summarization does not involve selecting from a fixed label set. Instead, we use $r_{\text{alignment}}$, which computes the reward based on the average of the three ROUGE metrics.

The results, shown in Table 2, indicate that PRL significantly outperforms all baseline methods on the summarization task. Interestingly, PRL consistently opts to generate prompts without incorporating few-shot examples.

We include the generated prompts in Figure 4, along with the corresponding average ROUGE scores. Notably, the two prompts produced by EvoPrompt are nearly identical in structure and wording, yet they yield significantly different results. This underscores how seemingly minor variations in prompt phrasing that are semantically equivalent to humans can lead to substantial differences in LLM performance.

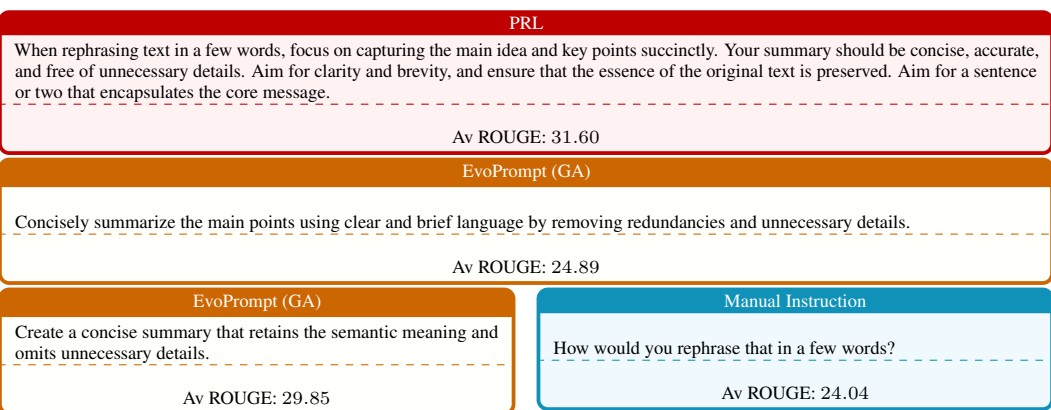

Figure 4: Comparison of averaged ROUGE metrics based on prompts generated by PRL, EvoPrompt, and Manual Instruction for the summarization task. This figure highlights the importance of precise prompt design: although the two prompts generated by EvoPrompt on two different seeds are superficially similar, they result in significantly different performance. In contrast, the PRL prompt is both more effective and better aligned with the task objective.

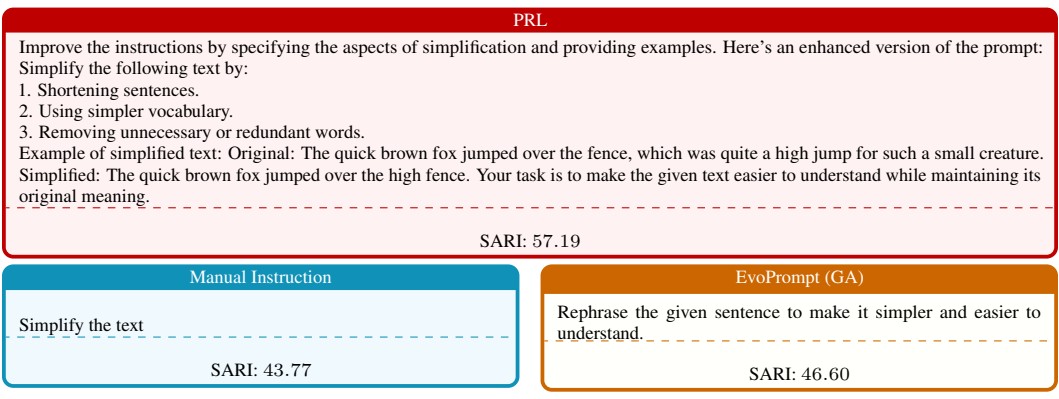

Figure 5: Comparison of SARI metric for prompts generated by PRL, EvoPrompt and Manual Instruction for the simplification task.

**Simplification**  We evaluate PRL on the sentence simplification task using the ASSET dataset (Alva-Manchego et al., 2020). ASSET is a crowdsourced corpus designed to assess the performance of simplification models across multiple rewriting operations, including lexical paraphrasing, sentence splitting, deletion, and reordering. Each of the original sentences is accompanied by human-written

simplifications, providing a rich set of references for evaluating model outputs. This multi-reference setup enables more robust and comprehensive evaluation of simplification systems.

To evaluate simplification quality, we adopt the SARI metric Xu et al. (2016), which compares the system output against both the original sentence and a set of reference simplifications. SARI assesses the quality of words that are added, deleted, and kept by the system. It has been shown to correlate well with human judgments of simplicity.

We set $r_{\text{format}} = 0$ for this task, as there is no fixed output format to enforce. For the alignment reward, we use the SARI metric. For the final scoring function, we also use the SARI score.

Table 3: Task simplification results.

The results are presented in Table 3. Our baselines perform on average comparable to a manually written prompt. Generated prompts from PRL, EvoPrompt, and Manual Instruction are provided in Figure 5. For this task baselines generated comparatively simple prompts which fail to provide sufficient guide the model's output. In contrast, the PRL prompt is precise, comprehensive, includes a well-constructed example and leads to a substantial performance improvement.

| Method | SARI |
|--------|------|
| MI | 43.77 |
| APE | $45.33_{\pm 0.83}$ |
| GA | $46.25_{\pm 0.47}$ |
| DE | $45.79_{\pm 0.35}$ |
| PRL | $52.26_{\pm 3.51}$ |

**GSM8K**  To ascertain robustness of our method w.r.t. other types of problems, we have tested PRL and the other baselines on the mathematical reasoning and problem-solving benchmark dataset GSM8K Cobbe et al. (2021). The experiment, including training, was performed under two evaluation protocols: (1) the answer is considered correct only when the LLM output matches exactly the correct integer, and (2) the answer is considered correct if the correct integer appears anywhere in the LLM output. The second protocol allows the evaluator model to engage in intermediate reasoning steps. We see in Table 4 that our method generalizes across problem domains yielding also SoTA results for reasoning based problems.

Interestingly, across all of our experiments, the results generally differ from those reported in the original EvoPrompt paper, which claimed that the differential evolution (DE) variant of prompt generation outperforms the genetic algorithm (GA) variant. In contrast, across all four types of tasks we observe that the GA variant consistently yields superior accuracy compared to DE. We attribute this discrepancy to the use of different underlying language models in our reproduction study. These findings suggest that the relative effectiveness of EvoPrompt's evolutionary strategies is sensitive to the choice of the base model.

**Ablation Study: Influence of Prompt Selection**  We analyze the impact of the Prompt Selection process on overall performance. In the ablation setting, instead of selecting the best prompt iteratively during training, we simply report the final prompt at the end of training. To do so after training we sample $n_{\text{test}}$ prompts to choose the best one according to the validation set. This comparison is performed across all classification tasks.

Table 4: Results on GSM8K. Superscripts [1] and [2] indicate the evaluation protocol.

| Method | Accuracy[1] | Accuracy[2] |
|--------|-------------|-------------|
| MI | 22.20 | 78.20 |
| APE | $27.17_{\pm 0.65}$ | $83.43_{\pm 1.98}$ |
| GA | $26.38_{\pm 1.10}$ | $81.62_{\pm 1.38}$ |
| DE | $26.38_{\pm 1.10}$ | $79.52_{\pm 0.45}$ |
| PRL | $29.30_{\pm 0.05}$ | $86.15_{\pm 0.55}$ |

The results, shown in Table 1, demonstrate that Prompt Sampling not only improves final performance but also enhances training efficiency. By selecting strong prompts throughout the training process, Prompt Selection leads to both better and faster results. We believe this phenomenon arises from two main factors. First, the use of reinforcement learning to train our LLMs, which always has significant variance. Second, our method is vulnerable to overfitting due to the limited number of samples in the training dataset.

**Ablation Study: Influence of Reasoning**  To investigate the role of explicit reasoning in our method, we conduct an ablation study based on the prompt design illustrated in Appendix B. In the standard setup, the model is instructed to perform a reasoning process before producing the final answer. To evaluate the effect of removing this step, we modify the prompt to omit the reasoning phase and instead directly request the model to generate the answer within `<answer> </answer>` tokens.

We perform this experiment on the SUBJ dataset, training two identical models, except for either using or or omitting reasoning. This difference leads to a substantial drop in accuracy, from 75.05 (±1.63) with reasoning to 60.12 (±1.62) without reasoning, showing the importance of explicit reasoning in our approach.

**Ablation Study: PRL on Larger Models**   It is commonly observed that larger, more powerful models are less sensitive to prompt variations. To investigate this phenomenon, we use Qwen2-{7B|14B|32B}-Instruct as the Evaluation Models, while keeping Qwen2-7B-Instruct as the Prompt Generator. For this experiment we utilize 6 A100 GPUs for the 32B model and 4 A100 GPUs for the 14B model.

We compare the performance using the base prompt against prompts generated by PRL on the MR dataset. The results, presented in Table 5, show that all model sizes benefit from PRL, demonstrating two key findings: (i) Even larger LLMs remain vulnerable to prompt variation. (ii) PRL is capable of effectively tailoring prompts for both smaller and larger models, significantly improving their performance.

Table 5: Comparison of accuracy across different model sizes of the Evaluation Model on the MR dataset.

| Parameters | 7B | 14B | 32B |
|---|---|---|---|
| MI | 87.40 | 89.20 | 90.15 |
| PRL | $91.27_{\pm0.05}$ | $92.03_{\pm0.13}$ | $92.52_{\pm0.02}$ |

**Ablation Study: PRL Beyond Qwen**   We test the cross model robustness of PRL using LLaMA 3.1-8B- Instruct AI@Meta (2024). First, we assess prompt portability: prompts learned by PRL and benchmark methods with Qwen2.5-7B- Instruct are applied unchanged to LLaMA-3.1-8B Instruct for summarization. As shown in Table 6, these prompts transfer well and remain competitive with strong baselines, indicating that PRL produces prompts that generalize beyond the backbone used to train them. Second, motivated by observations of potential spurious gains when training with Qwen under weak reward signals Shao et al. (2025), we retrain the prompt generator on LLaMA-3.1-8B Instruct (reported as PRL-LLaMA in Table 6). The resulting prompts achieve equal or better ROUGE 1/2 than the Qwen trained counterpart. Together, these findings suggest that PRL's improvements are not an artifact of a particular model family: PRL trained prompts both transfer across architectures and train effectively on alternative backbones.

We present two additional ablation study experiments in Appendix D.

## 5   CONCLUSIONS & LIMITATIONS

We have introduced an RL-based algorithm for prompt generation that consistently outperforms other approaches across classification, summarization and simplification tasks.

Even though we use recent LLMs, better prompts can still significantly increase task performance, indicating that LLMs are still sensitive to differences in semantically equivalent prompts. Interestingly, this holds true even for the largest LLM we have tested on, the Qwen2-32B-instruct model. Additionally, our results underscore that there is no single recipe to generate good prompts across different tasks, as some tasks benefit from few-shot examples or other subtle semantic cues, while others do not. Our approach effectively navigates

Table 6: Task summarization results evaluated using LLaMA.

| Method | ROUGE-1 | ROUGE-2 | ROUGE-L |
|---|---|---|---|
| MI | 33.33 | 10.77 | 27.12 |
| APE | $34.12_{\pm3.86}$ | $12.90_{\pm2.56}$ | $25.72_{\pm3.63}$ |
| GA | $38.73_{\pm2.36}$ | $14.69_{\pm1.01}$ | $30.38_{\pm2.94}$ |
| DE | $34.25_{\pm3.56}$ | $11.57_{\pm2.86}$ | $25.43_{\pm3.65}$ |
| PRL | $39.38_{\pm2.82}$ | $15.77_{\pm1.56}$ | $30.57_{\pm2.80}$ |
| PRL-LLaMA | $43.70_{\pm0.02}$ | $16.66_{\pm0.25}$ | $37.46_{\pm0.5}$ |

such delicate prompt crafting issues. In line with current work our prompt generator profits from increased inference time compute by allowing it to reason about effective prompts.

Currently, improved performance is obtained via a significantly greater computational expense than used by the comparatively simpler related work. Another limitation is that we retrain the prompt generator for each new task. A universal prompt generator is a desideratum.

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

# A  APPENDIX - PRL PSEUDO CODE

In Algorithm 1 we provide the PRL algorithm.

---

**Algorithm 1** PRL

---

**Require:** $\pi_\theta^{\text{generator}}$: prompt generator
    $\pi^{\text{eval}}$: frozen Evaluation Model
    $T, V$: training and validation datasets
    $n, n_{\text{test}}$: number of prompts during training/Prompt Selection
    $k$: number of samples per iteration
    $I$: total number of iterations
    $t$: Prompt Selection frequency
    $R$: reward operator
    $f$: scoring function

1:  $best\_score \leftarrow 0, best\_prompt \leftarrow$ ""
2:  **for** $i = 1$ to $I$ **do**
3:     Sample $k$ training samples $D \sim T$
4:     Generate answers $o_1, \ldots, o_n \sim \pi_\theta^{\text{generator}}$
5:     Extract answers $p_1, \ldots, p_n$ from $o_1, \ldots, o_n$
6:     Compute rewards $r_j = R(\pi^{\text{eval}}, D, p_j, o_j)$ for each $j = 1, ..., n$
7:     Update $\theta$ using GRPO with rewards $\{r_j\}$
8:     **if** $i \bmod t = 0$ **then**
9:       Generate test prompts $p_1, \ldots, p_{n_{\text{test}}} \sim \pi_\theta^{\text{generator}}$
10:       **with** `torch.no_grad()`:
11:       Compute scores $s_j = f(\pi^{\text{eval}}, V, p_j)$
12:       Let $j^* = \arg\max_j s_j$
13:       **if** $s_{j^*} > best\_score$ **then**
14:         $best\_score \leftarrow s_{j^*}, best\_prompt \leftarrow p_{j^*}$
15:       **end if**
16:     **end if**
17: **end for**
18: **return** $best\_prompt$

---

# B APPENDIX - PRL PROMPT FORMAT

We present below the prompt format used by PRL. In the system prompt, we instruct the model to generate a reasoning trace enclosed within <think> and </think> tokens, followed by the final answer encapsulated within <answer> and </answer> tokens. The user message provides the base prompt that it should refine. The model's objective is to produce the prompt that is better than the best prompt.

| System Prompt | User Prompt |
|---|---|
| A conversation between User and Assistant. The user asks a question, and the Assistant solves it. The assistant first thinks about the reasoning process in the mind and then provides the user with the answer. The reasoning process and answer are enclosed within <think> </think> and <answer> </answer> tags, respectively, i.e., <think> reasoning process here </think><answer> answer here </answer> | Your task is to refine a base prompt for another model that performs a sentiment classification task. Improve the instructions to enhance the model's performance. The base prompt: In this task, you are given sentences from movie reviews. The task is to classify a sentence as 'positive' if the sentiment of the sentence is positive or as 'negative' if the sentiment of the sentence is negative. Return label 'positive' or 'negative' only without any other text. |

Figure 6: Prompt used by PRL

# C APPENDIX – PROMPTS FOR CLASSIFICATION

This subsection provides the most effective prompts used for the classification task in our method.

| PRL |
|---|
| In this task, you are to classify the opinion in a given sentence from a review as either subjective or objective. |
| - A subjective sentence expresses personal feelings, opinions, or attitudes. |
| - An objective sentence presents facts that can be verified and are not influenced by personal feelings. |
| Examples: |
| - Subjective: "This movie is the best I've ever seen." (Opinion expressed) |
| - Objective: "The movie won five awards this year." (Fact stated) |
| When classifying, focus only on the opinion, not the facts. Return the label 'subjective' or 'objective' only, without any additional text. |
| Example: |
| Input: "The food was delicious." Output: subjective |
| Acc.: 77.95 |

Figure 7: Best prompt generated by PRL for SUBJ classification task along with accuracy.

| PRL |
|---|
| In this task, you will classify the sentiment of movie review sentences as 'positive' or 'negative'. Examples: |
| "The movie was thrilling and exciting" -> positive; |
| "The plot was boring and predictable" -> negative. |
| Return only the label: 'positive' or 'negative'. |
| Acc.: 96.38 |

Figure 8: Best prompt generated by PRL for SST2 classification task along with accuracy.

**PRL**

In this task, you are given sentences from movie reviews. Your goal is to classify each sentence as 'positive' or 'negative' based on its sentiment. Pay close attention to the context and nuances in the text, as the sentiment might not be explicitly stated. Examples:
- "The acting was superb, and the plot was engaging." -> positive
- "The movie was so slow and boring that I almost fell asleep." -> negative
Return only the label 'positive' or 'negative' without any additional text.

Acc.: 93.00

Figure 9: Best prompt generated by PRL for CR classification task along with accuracy.

**PRL**

In this task, you are given a sentence from a movie review. Classify the sentence as 'positive' if the sentiment is positive, or as 'negative' if the sentiment is negative. Provide only the label 'positive' or 'negative' without any additional text. Examples:
- "The acting was superb and the plot was engaging." -> positive
- "The movie was boring and the storyline was predictable." -> negative

Acc.: 91.30

Figure 10: Best prompt generated by PRL for MR classification task along with accuracy.

**PRL**

In this task, you are given sentences from movie reviews. Your goal is to classify the sentiment of each sentence as 'terrible', 'bad', 'okay', 'good', or 'great'. Be as accurate as possible. Here are the guidelines for each category:
- 'terrible': The sentence expresses extreme dissatisfaction or negative feelings.
- 'bad': The sentence conveys negative feelings but not as strongly as 'terrible'.
- 'okay': The sentence is neutral or has mixed feelings with no strong positive or negative sentiment.
- 'good': The sentence conveys positive feelings but not as strongly as 'great'.
- 'great': The sentence expresses strong positive feelings or high satisfaction.
Consider the overall tone and specific positive or negative words in the sentence to determine the closest sentiment.
If you are not sure, choose the closest option.
Return the label 'terrible', 'bad', 'okay', 'good', or 'great' only without any additional text.

Acc.: 56.38

Figure 11: Best prompt generated by PRL for SST-5 classification task along with accuracy.

**PRL**

In this task, you will be given a news article and asked to classify it into one of the four predefined categories: 'World', 'Sports', 'Business', or 'Tech'.
Follow these detailed instructions to ensure accurate classification
1. Read the article thoroughly to understand its main subject matter.
2. Determine which of the following categories the article's main topic most closely aligns with:
- 'World': articles covering global news, politics, international affairs, etc.
- 'Sports': articles discussing various sports, competitions, athletes, etc.
- 'Business': articles focusing on financial news, corporate activities, markets, etc.
- 'Tech': articles about technology, innovations, companies, gadgets, etc.
3. If the article's content is not clearly related to any of these categories, choose the closest option based on the predominant subject matter.
4. Return the label of the chosen category as a single word without any additional text or explanations, e.g., 'World', 'Sports', 'Business', or 'Tech'.
Example:
Article: "Apple Launches New iPhone Model with Improved Camera Features" Label: Tech
Article: "China and the US Reach a New Trade Agreement" Label: World
Article: "Local Soccer Team Qualifies for the World Cup" Label: Sports
Article: "Oil Prices Drop as OPEC Decides to Cut Production" Label: Business

Acc.: 84.42

Figure 12: Best prompt generated by PRL for AG's News classification task along with accuracy.

Figure 13: Best prompt generated by PRL for TREC classification task along with accuracy.

# D APPENDIX - ADDITIONAL ABLATION STUDY EXPERIMENTS

**Impact of Prompt Sampling Size in Prompt Selection** We investigate how the number of prompt samples used during inference in the Prompt Selection technique affects final performance. Specifically, we evaluate PRL on the MR classification dataset while varying the number of sampled prompts: $n_{\text{test}} = 1, 5, 10$, and $15$. Each configuration is run three times, and we report the average accuracy. The results are presented in Table 7.

Performance remains stable when using more than five prompt samples, while one prompt only leads to a performance drop. Although the results suggest that five prompts are sufficient for stable performance, we recommend using ten prompts to provide an additional buffer against potential sensitivity in other tasks or datasets.

Table 7: Model accuracy vs. number of test samples

| $n_{\text{test}}$ | 1 | 5 | 10 | 15 |
|---|---|---|---|---|
| Accuracy | $90.92_{\pm 0.17}$ | $91.25_{\pm 0.15}$ | $91.27_{\pm 0.11}$ | $91.35_{\pm 0.11}$ |

**Influence of Few-Shot Examples** To ascertain the importance of the automatic inclusion of few-shot examples, we manually remove them from prompt for which PRL provided them and measure performance, see Table 8 for results on a subset of classification tasks where PRL produced few-shot examples. We see that indeed few-shot examples significantly enhance quality.

Table 8: Accuracy on different datasets with and without few-shot learning.

| Dataset | Acc. w/o few-shot | Acc. with few-shot |
|---|---|---|
| **SUBJ** | 66.75 | 77.95 |
| **CR** | 92.40 | 93.00 |
| **SST-2** | 95.00 | 96.38 |
| **MR** | 90.95 | 91.30 |

# E APPENDIX – BENCHMARK RESULTS REPRODUCTION

To provide a fair comparison with existing benchmarks (APE, APO, and EvoPrompt), we reproduced their results using Qwen2.5-7B-Instruct as both the Prompt Generator and the Evaluation Model, terms named differently in the original papers but are functionally equivalent. The evaluation procedure is identical for all the benchmarks and for PRL, as follows:

- For classification tasks, only a response consisting of a single word denoting the correct class is considered a valid answer.
- For summarization and simplification tasks, the entire response generated by the Evaluation Model is used to compute ROUGE/SARI metrics.

- For GSM8K task, a response including only the proper integer is considered a valid answer.

To ensure that only the label is output by the Evaluation Model, we appended the following instruction to the end of each prompt in the initial population:

```
Return only label {list of labels} without any other
text.
```

For example, in the case of the SST-5 dataset, the appended sentence was:

```
Return only label 'terrible', 'bad', 'okay', 'good' or
'great' without any other text.
```

In GSM8K task, the instruction to output only correct integer is added to the initial prompt but only when using the evaluation protocol (1), see paragraph GSM8K.

Depending on the benchmark, the initial population consisted of manual prompts (see MI in Baseline Methods) and/or automatic prompts generated by Qwen, following the prompt generation method described in (Zhou et al., 2022).

**APO**  Following the EvoPrompt (Guo et al., 2023) experimental protocol, we ran the APO algorithm using the manual prompt with the best performance as the initial population. However, unlike the EvoPrompt authors, we applied the method to all classification tasks, not only binary ones. APO was not evaluated on text generation tasks (i.e., summarization and simplification), as its optimization algorithm fundamentally relies on binary feedback (correct vs incorrect), which is incompatible with continuous scores such as ROUGE or SARI.

The default parameter setup provided by (Pryzant et al., 2023) was used for each run.

**APE and EvoPrompt**  Following (Guo et al., 2023), the development set size was set to 200 for classification tasks and 100 for simplification and summarization tasks. Each run included 10 iterations. The 10 best prompts for each task served as the initial population (selected from automatic prompts for APE and from both automatic and manual prompts for EvoPrompt).

**RLPrompt**  A direct adaptation of RLPrompt (Deng et al., 2022) to PRL setting was not feasible, since it relies on a fundamentally different evaluation paradigm (selecting the token with the highest probability from a list of predefined verbalizers), much smaller training datasets, and significantly smaller language models. Instead of retraining RLPrompt on Qwen (which would be meaningless in our opinion), we compared the prompts generated by PRL with those reported in the RLPrompt paper, using RLPrompt's own evaluation method. To ensure fairness despite differences in prompt templates, we evaluated each method with three variants:

- sample + prompt; without chat template,
- sample + prompt; with chat template,
- prompt + sample; with chat template

and selected the best score. This approach is justified by the RLPrompt authors' claim that its prompts exhibit inter-model generalizability. Under this setup, PRL surpasses RLPrompt on every task except SUBJ and achieves a higher average score by 16.5% (see Table 9).

Table 9: Accuracy achieved by prompts from RLPrompt and PRL on classification tasks. Red colour mark the best result. The right-most column shows the mean accuracy of each method across the seven datasets.

| Method / Dataset | SST-2 | CR | MR | SST-5 | AG's News | TREC | Subj | Avg |
|---|---|---|---|---|---|---|---|---|
| RLPrompt (2 tokens) | 65,79 | 58.65 | 72.45 | 40.23 | 70.42 | 38.40 | 68.90 | 59.26 |
| RLPrompt (5 tokens) | 82.98 | 76.20 | 82.25 | 27.24 | 65.07 | 39.60 | 74.10 | 63.92 |
| PRL (ours) | 95.55 | 88.60 | 91.85 | 56.02 | 87.39 | 76.40 | 67.10 | 80.42 |

## F   APPENDIX – DISCLOSING STATEMENT ON THE USE OF LARGE LANGUAGE MODELS

LLMs were used solely as an assistive tool for text formatting and minor language refinement. They did not contribute to research ideation, experimental design or analysis. The authors take full responsibility for the final content of this paper.

