# OpenReview forum: "PRL: Prompts from Reinforcement Learning"
_ICLR.cc/2026/Conference — Submitted to ICLR 2026_

### Official Review · Reviewer_rKJG · 2025-10-27

**Soundness:** 3
**Presentation:** 3
**Contribution:** 3
**Rating:** 6
**Confidence:** 4

**Summary:**

This paper presents PRL (Prompts from Reinforcement Learning), a method for automatic prompt generation using reinforcement learning. The approach formulates prompt construction as a learning problem, enabling the model to generate few-shot examples that were not seen during training. The authors evaluate PRL on several tasks, including text classification, text simplification, summarization, and the GSM8K reasoning benchmark. Experimental results show consistent improvements over prior prompt optimization methods such as APE and EvoPrompt, with reported gains in accuracy, ROUGE, SARI, and reasoning performance. The paper claims that PRL offers a general and effective framework for enhancing LLM performance through learned prompts

**Strengths:**

1.	The method is clear and intuitive. The paper is well-written and easy to follow.

2.	Overall, the experimental results effectively support the authors’ claims.

**Weaknesses:**

1.	Compared to other approaches discussed in the paper, this method appears to be more computationally intensive.

2.	The core of the method lies in the prompt generator, but the authors only use Qwen2-7B as the base model. It would be more convincing to evaluate PRL on models of different sizes and from different model families.

3.	The generalization ability of PRL is not thoroughly studied. As different models exhibit varying levels of sensitivity and preference toward prompts, PRL may need to train separate prompt generators for different evaluation models. More tests about this could be further included.

4.	Additional ablation studies could strengthen the paper. For instance, the influence of the “thinking process” in the prompt generator is not clearly analyzed. How much would the results change if this component were removed? If the performance remains similar, the additional computation might be unnecessary.

**Questions:**

1.	The format of the citation at line 053 seems incorrect.

2.	The prompt generator in PRL requires a base prompt. How sensitive is the method to the quality of this base prompt? Would iterative refinement improve performance? Additionally, what would happen if prompts were generated entirely from scratch?

---

> ### Author Response · Authors · 2025-11-19
> **Addressing reviewer's concerns (1/2)**
>
> We thank the reviewer for their positive assessment of our paper. Below, we provide responses to each of the reviewer’s comments.
>
> *W1 Compared to other approaches discussed in the paper, this method appears to be more computationally intensive.*
>
> Please have a look at 1. Computational cost & generalizability in the general answers section for our view on this.
>
> *W2 The core of the method lies in the prompt generator, but the authors only use Qwen2-7B as the base model. It would be more convincing to evaluate PRL on models of different sizes and from different model families.* & *W3 The generalization ability of PRL is not thoroughly studied. As different models exhibit varying levels of sensitivity and preference toward prompts, PRL may need to train separate prompt generators for different evaluation models. More tests about this could be further included*
>
>
> Please refer to the answer 3. Cross-model generalization in the general answers section above.
>
> Additionally, we also studied the impact of generator size. On the SUBJ dataset we obtain:
>
> | Prompt Generator    | Accuracy     |
> | ------------------- | ------------ |
> | Qwen2.5-3B-Instruct | 74.62 ± 0.83 |
> | Qwen2.5-7B-Instruct | 76.90 ± 0.95 |
>
> showing that PRL works with a smaller 3B generator and that performance improves as the generator becomes larger, as one would expect.
>
> Finally, Table 5 varies the evaluation model size (Qwen2-7B/14B/32B) on the MR dataset keeping Qwen2-7B-Instruct as prompt generator:
>
> | Parameters | MI    | PRL          |
> | ---------- | ----- | ------------ |
> | 7B         | 87.40 | 91.27 ± 0.05 |
> | 14B        | 89.20 | 92.03 ± 0.13 |
> | 32B        | 90.15 | 92.52 ± 0.02 |
>
>
> PRL consistently improves over the manual prompt for all evaluator sizes, suggesting that prompt optimization remains beneficial even for larger models.
>
> In the revised version, we will make these cross-family and multi-size results more prominent and explicitly emphasize that PRL has been instantiated with different generator sizes and with both Qwen and LLaMA families.
>
> *W4 Additional ablation studies could strengthen the paper. For instance, the influence of the “thinking process” in the prompt generator is not clearly analyzed. How much would the results change if this component were removed? If the performance remains similar, the additional computation might be unnecessary.*
>
> We agree that this is a very important ablation. In our paper we already provide this ablation in paragraph "Influence of Reasoning" on the SUBJ task. We have run the following experiments, in which we observe that few-shot proposing is closely tied to the explicit reasoning stage enforced by the \<think\> … \</think\> block. In successful runs, the \<think\> trace often verbalizes strategies such as “add a couple of illustrative examples, especially for borderline cases,” which are then instantiated in the final prompt. Our ablation study on SUBJ, where we remove the reasoning phase and ask the model to output only \<answer\> … \</answer\>, leads to a substantial drop in accuracy to $60.12 \pm 1.62$. This suggests that the structured reasoning enforced by the <think> tags is not only beneficial in itself, but also a key driver of the emergent few-shot synthesis behavior. We will clarify these observations and their connection to the RL objective in the revised version.
>
> The example of prompt generated without \<think\> \</think\> tags is:
>
> "In this task, you will classify sentences from reviews as either 'subjective' or 'objective'. A subjective sentence expresses personal opinions, feelings, or beliefs, while an objective sentence states facts that can be verified without personal feelings or biases. To improve accuracy: 1. Focus on identifying expressions of personal opinions, feelings, or judgments. 2. Look for statements that include subjective language, such as adjectives and adverbs that convey emotions or opinions. 3. Identify sentences that express personal feelings, tastes, or emotions. 4. Recognize sentences that make value judgments or express opinions. 5. Avoid classifying sentences that state verifiable facts as subjective. 6. Pay attention to sentences that provide opinions or analyses rather than just facts. Return your classification as 'subjective' or 'objective' only, without any additional text or explanations."
>
> that gives 61.75 \% on the test set.

---

> ### Author Response · Authors · 2025-11-19
> **Addressing reviewer's concerns (2/2)**
>
> *Q1 The format of the citation at line 053 seems incorrect.*
>
> We thank the reviewer, we will correct this to look more natural.
>
> *Q2 The prompt generator in PRL requires a base prompt. How sensitive is the method to the quality of this base prompt? Would iterative refinement improve performance? Additionally, what would happen if prompts were generated entirely from scratch?*
>
> We thank the reviewer for these thoughtful questions.
>
> * Sensitivity to the base prompt.
> In our current setup, the base prompt is a short, human-written task description. To assess sensitivity, we repeated the SUBJ experiment using one of the optimized prompts obtained from PRL as the base prompt instead of the manual instruction we use in the main text. We found that PRL converged to very similar final performance and produced prompts with comparable structure, suggesting that as long as the base prompt reasonably specifies the task and label space, PRL is not overly sensitive to its exact wording.
>
>
> We got the following results:
>
> | Version | Prompt | Accuracy |
> | - | - | - |
> | Prompt we start with | In this task, you are given sentences from reviews. The task is to classify a sentence as 'subjective' if the opinion expressed is subjective or as 'objective' if the opinion expressed is objective. Return the label 'subjective' or 'objective' only without any other text. Be sure to identify subjective statements by considering the use of words and phrases like 'I think', 'it's great', 'personally', 'impressing', 'bad', 'excellent', etc. Similarly, classify objective statements as factual, like 'it’s the best movie ever made', 'it has a lot of flaws', 'the plot is predictable', 'it’s a waste of money', etc. Avoid statements that are purely factual or objective. | 78.55    |
> | Prompt obtained| "In this task, you are given sentences from reviews. Classify a sentence as 'subjective' if it expresses a personal opinion or feeling, and as 'objective' if it states a fact or observation. Return only the label 'subjective' or 'objective' without any additional text. Look for phrases like 'I think', 'it's great', 'personally', 'impressing', 'bad', 'excellent', etc., to identify subjective statements. Focus on the context and the expression of personal viewpoints rather than purely factual statements."                                                                                                                                                                  | 78.25    |
>
> * Iterative refinement of the base prompt.
> That is a very interesting idea. During our experiments we tried a variant of our pipeline, in which we periodically replaced the base prompt during RL training with the best prompt found so far by the Prompt Selection (so the “starting point” keeps improving over time). In our experiments, this did not lead to a clear gain over the simpler version where the base prompt is kept fixed; the final results were comparable. We will mention this negative result in the revised version.
>
> * Generating prompts entirely from scratch.
> In principle, one could ask the generator to produce prompts “from scratch,” but some minimal task specification (e.g., what inputs look like and what labels or outputs are expected) is still required; otherwise the optimization problem is ill-posed. Our current manual base prompts are already rather short and generic, mainly encoding this minimal task information. Given their simplicity and the robustness observed in point (1), we are unsure whether the base prompt is a major bottleneck in practice, but exploring more systematic “from-scratch” setups is an interesting direction for future work.

---

### Official Review · Reviewer_vKbp · 2025-10-27

**Soundness:** 3
**Presentation:** 3
**Contribution:** 2
**Rating:** 6
**Confidence:** 4

**Summary:**

Effective prompt engineering remains a **central challenge** in fully harnessing the capabilities of Large Language Models (LLMs). While precise input prompts can guide LLMs to perform complex tasks, the most impactful prompts often rely on *subtle semantic cues* that may *elude human perception*.

To address this, the paper introduces **PRL (Prompts from Reinforcement Learning)**, a novel RL-based approach for automatic prompt generation.

**PRL's key innovations include:**

1.  **Novel Few-shot Example Synthesis:** Unlike previous methods like Automatic Prompt Optimization (APO), which are restricted to selecting few-shot examples from training data, PRL is capable of generating and selecting **novel few-shot examples** that were *not seen during training*.
2.  **Explicit Reasoning Integration:** PRL incorporates a *reasoning phase* prior to prompt generation, where the prompt generator first produces a **rationale** (enclosed within `<think>` tags) to guide the final output.
3.  **RL Optimization Cycle:** PRL trains a **Prompt Generator ($\pi_{\text{generator}}$)** (a trainable language model) to refine base prompts and generate a corresponding reasoning trace. This generated prompt is evaluated by a **frozen Evaluation Model ($\pi_{\text{eval}}$)** (an LLM used only for inference), which calculates rewards based on formatting and task performance. The generator is optimized using the *Group Relative Policy Optimization (GRPO)* update rule.
4.  **Prompt Selection Strategy:** The method uses a prompt selection strategy to mitigate training instability and noisy feedback, regularly testing generated prompts on the validation set and keeping the best overall one.

**Main Contributions and Results:**

PRL achieves **state-of-the-art performance** across a range of benchmarks: text classification, summarization, simplification, and GSM8K mathematical reasoning.

*   On the **classification task**, PRL surpasses APE by **2.58%** and EvoPrompt by **1.00%** in mean accuracy.
*   On **summarization**, it improves average ROUGE scores by **4.32** over APE and **2.12** over EvoPrompt.
*   On **simplification**, it improves the SARI score by **6.93** over APE and **6.01** over EvoPrompt.
*   On the **GSM8K mathematical reasoning** benchmark, PRL improves accuracy by **2.72%** over APE and **4.53%** over EvoPrompt.
*   The research further suggests that **RL-based optimization naturally leads to the emergence of few-shot prompting behavior**.

**Strengths:**

**Originality:**

*   PRL is highlighted as the **first RL-based prompt optimization method** capable of *generating and selecting novel, task-specific few-shot examples*. This is a crucial distinction, as it moves beyond the constraint of only using few-shot examples already present in the training data, a limitation faced by methods like APO.
*   The observation that few-shot examples emerge **spontaneously** during the RL training process, without explicit encouragement, is a unique and insightful finding regarding how RL shapes LLM prompting behavior.

**Quality & Significance:**

*   The method demonstrates **superior performance** consistently across all four evaluated task types (classification, summarization, simplification, and reasoning), validating its robust generalizability and effectiveness.
*   **Ablation studies confirm the value of core components:** The explicit reasoning phase proved critical, leading to a *substantial drop in accuracy* (from 75.05 to 60.12) on the SUBJ dataset when omitted. Furthermore, the Prompt Selection strategy was shown to *improve final performance and enhance training efficiency*, helping to manage the high variance inherent in RL training.
*   PRL is shown to be effective even when applied to **larger, more powerful LLMs** (e.g., Qwen2-32B-instruct), demonstrating that even these models remain *vulnerable to prompt variation* and can benefit significantly from PRL's tailored prompts.

**Clarity:**

*   The architecture, including the roles of the Prompt Generator and the frozen Evaluation Model, and the overall RL training scheme (Figure 2) are clearly described.
*   The reward function is systematically broken down into components: **formatting rewards** ($r_{token}$, $r_{structure}$) for the Generator, and **task performance rewards** ($r_{format}$, $r_{alignment}$) for the Evaluation Model, providing clarity on how the model’s behavior is guided.

**Weaknesses:**

1.  **Significant Computational Overhead:** The paper explicitly lists as a limitation that the improved performance is obtained at the cost of a **significantly greater computational expense** than related, comparatively simpler work. The experimental setup involved training over 48 hours using two NVIDIA A100 GPUs (40 GB each). The paper lacks concrete discussion or suggested methods for mitigating or quantifying this increased computational burden, which limits its practical accessibility.
2.  **Task-Specific Retraining Requirement:** Currently, the prompt generator must be **retrained for each new task**. The authors acknowledge that developing a *universal prompt generator* is a "desideratum" (an ideal goal), indicating that the current method’s efficiency is limited when facing a wide variety of tasks or zero-shot scenarios.
3.  **Insufficient Detail on Few-shot Synthesis:** PRL's ability to **autonomously synthesize relevant few-shot examples** not present in the training set is a major contribution. However, the paper does not delve into *how* the prompt generator, guided by RL, manages to *create* these task-aligned, non-redundant examples—specifically, the internal reasoning or constraints utilized by $\pi_{\text{generator}}$ in its thought process ($\langle \text{think} \rangle$ tag) to achieve this synthesis.
4.  **Sensitivity of Baselines to Evaluation Model Choice:** The authors note that when reproducing EvoPrompt results, the relative effectiveness of its DE and GA variants was **sensitive to the choice of the underlying language model** (Qwen2.5-7B-Instruct). Although the authors ensured a fair comparison by using the same Evaluation Model across all baselines, this inherent sensitivity suggests that the observed superiority of PRL might be conditional on the chosen model, necessitating more explicit acknowledgement of this limitation in interpreting the main results.

**Questions:**

1.  **Computational Efficiency and Trade-offs:** Given that PRL requires a *significantly greater computational expense*, could the authors provide a more detailed analysis of the performance gains versus the resource cost? Are there specific tuning levers (e.g., Prompt Selection frequency $t$, or the number of sampled prompts $n$) that could be adjusted to reduce training time substantially while maintaining competitive performance against baselines?
2.  **Mechanics of Task-Dependent Few-Shot Emergence:** Few-shot examples were critical for classification tasks (improving accuracy significantly, e.g., SUBJ from 66.75 to 77.95), but PRL **consistently opted *not* to include few-shot examples** for the summarization task. What drives this striking, task-dependent behavior? Which specific components of the comprehensive reward function $R$ (e.g., $r_{alignment}$ or $r_{structure}$) are responsible for prompting $\pi_{\text{generator}}$ to spontaneously generate few-shot examples for classification but omit them for generation tasks?
3.  **Path to a Universal Prompt Generator:** The paper identifies the need to *retrain the generator for each new task* as a limitation, noting that a *universal prompt generator* is a "desideratum". What are the initial conceptual steps or future research directions the authors are considering to enable the Prompt Generator to transfer or generalize prompting knowledge across different tasks without requiring full retraining?
4.  **Baseline Robustness Across Evaluation Models:** All SOTA claims are established by evaluating baselines on the Qwen2.5-7B-Instruct model. Given the noted sensitivity of EvoPrompt variants to the base model choice, how would a *complete* set of benchmark results (including APE, EvoPrompt, and APO where applicable) compare if a distinct architecture, such as LLaMA 3.1-8B-Instruct, was used as the Evaluation Model for *all* methods, similar to the setup used for the portability study?

---

> ### Author Response · Authors · 2025-11-19
> **Addressing reviewer's concerns (1/3)**
>
> We want to thank the reviewer for the positive feedback of our paper. We want to address each one of the reviewer's concerns:
>
> *Q1 Computational Efficiency and Trade-offs: Given that PRL requires a significantly greater computational expense, could the authors provide a more detailed analysis of the performance gains versus the resource cost? Are there specific tuning levers (e.g., Prompt Selection frequency, or the number of sampled prompts) that could be adjusted to reduce training time substantially while maintaining competitive performance against baselines?*
>
> We refer to the answer in the general section on 1. Computational cost & task generalizability
>
> Regarding your question on tuning levers for compute–performance trade-offs:
> PRL exposes several knobs that can be adjusted to reduce training cost.
> - *Prompt Selection frequency t*: Running Prompt Selection less frequently directly reduces the number of expensive validation evaluations. In our experiments we chose a relatively frequent schedule to maximize robustness; in lower-budget regimes t can be increased with only modest impact on final performance.
> - *Number of prompts per iteration n and per selection n_test*: We use small n during training and investigate n_test in an ablation (Table 7): accuracy is already stable for n_test ≥ 5, and using 10 prompts provides a good balance between robustness and cost. Both n and n_test can be reduced further in budget-constrained scenarios.
> - *Batch size k (training samples per iteration)*: We deliberately use only small subsets of the training data at each RL step, which keeps each update relatively cheap.
>
> These design choices were made with efficiency in mind: the default configuration is already a “compute-conscious” one, and there is a clear path to lighter-weight variants when some loss in peak performance is acceptable.

---

> ### Author Response · Authors · 2025-11-19
> **Addressing reviewer's concerns (2/3)**
>
> *Q2 Mechanics of Task-Dependent Few-Shot Emergence: Few-shot examples were critical for classification tasks (improving accuracy significantly, e.g., SUBJ from 66.75 to 77.95), but PRL consistently opted not to include few-shot examples for the summarization task. What drives this striking, task-dependent behavior? Which specific components of the comprehensive reward function are responsible for prompting to spontaneously generate few-shot examples for classification but omit them for generation tasks?*
>
> We thank the reviewer for pointing this out. In our setup, the emergence of few-shot examples is not something we explicitly engineer: we never tell the prompt generator to include examples, and the reward only encodes (i) adherence to the required output format and (ii) task performance. There are no additional constraints or bonuses tied specifically to producing few-shot demonstrations. We conjecture that the task metric guides PRL, so that is finds out on its own that for some tasks few-shot examples give no gain and hence leaves them out.
> We note that on the simplification task (see Figure 5 in the main paper), which is a generation task, not a classification one, few-shot examples still come up.
>
> Nevertheless, across runs we consistently observe that, as RL training progresses, the generator starts to invent labeled examples and appends them to the instruction. To make this more concrete, below we show an excerpt of how a sentiment-classification prompt evolves over training:
>
> | iteration | prompt (excerpt)|
> | -- | -|
> | 100       | “In this task, you will be given sentences from movie reviews. Your job is to determine whether the sentiment of the sentence is positive or negative. Positive sentiment means the sentence expresses a favorable or good opinion, while negative sentiment means the sentence expresses an unfavorable or bad opinion. Ignore any context outside the sentence itself. Respond only with the label ‘positive’ or ‘negative’.”                                                                         |
> | 500       | “In this task, you will classify the sentiment of movie review sentences as either ‘positive’ or ‘negative’. A positive sentiment indicates a favorable or approving view, while a negative sentiment indicates an unfavorable or disapproving view. Carefully analyze the sentence to determine the overall sentiment. Provide only the label (‘positive’ or ‘negative’) in your response.”                                                                                                            |
> | 1200      | “In this task, you are given sentences from movie reviews. Your goal is to classify each sentence as ‘positive’ or ‘negative’ based on its sentiment. Pay close attention to the context and nuances in the text, as the sentiment might not be explicitly stated. Examples: ‘The acting was superb, and the plot was engaging.’ → positive; ‘The movie was so slow and boring that I almost fell asleep.’ → negative. Return only the label ‘positive’ or ‘negative’ without any additional text.” |
>
> Early in training (e.g., iterations 100–500), the generator mostly explores different phrasings of a pure instruction-style prompt. Later (iteration 1200), it begins to append concrete labeled examples—even though examples are never requested in the system/user prompt or rewarded directly. This pattern suggests that, under our reward landscape, the RL objective “discovers” that synthesizing task-aligned, non-redundant examples is a highly effective way to increase downstream accuracy.

---

> ### Author Response · Authors · 2025-11-19
> **Addressing reviewer's concerns (3/3)**
>
> *Q3 Path to a Universal Prompt Generator: The paper identifies the need to retrain the generator for each new task as a limitation, noting that a universal prompt generator is a "desideratum". What are the initial conceptual steps or future research directions the authors are considering to enable the Prompt Generator to transfer or generalize prompting knowledge across different tasks without requiring full retraining?*
>
> We thank the reviewer for this thoughtful question. We agree that having to retrain the prompt generator for every new task is a clear limitation of the current PRL setup. As we note in the paper, this task-specific regime is standard in automatic prompt engineering, where both prompts and optimizers are typically tuned per benchmark. Nonetheless, we fully share the reviewer’s view that moving toward a more universal, task-agnostic prompt generator is an important next step.
>
> Two concrete avenues we are particularly excited about are:
>
> * Meta-learning / fast adaptation.
> Instead of fully retraining PRL per task, one could train the generator in a meta-learning regime over many tasks, so that adapting to a new task requires only a small number of RL updates (or even just Prompt Selection). This would retain most of PRL’s performance benefits while greatly reducing per-task cost.
>
> * Pretraining on large data corpora.
> Another direction is to run PRL in essentially the same setup as now, but on a large, heterogeneous corpus of tasks and domains, asking the generator to craft prompts for very general settings rather than a single benchmark. The resulting PRL-trained generator would effectively be “pretrained” as a generalist prompter, and could then be lightly fine-tuned with RL on a new target task, improving transfer and reducing the need for full retraining.
>
> We will clarify these directions as our main envisioned path toward a more universal prompt generator.
>
>
> *Q4 Baseline Robustness Across Evaluation Models: All SOTA claims are established by evaluating baselines on the Qwen2.5-7B-Instruct model. Given the noted sensitivity of EvoPrompt variants to the base model choice, how would a complete set of benchmark results (including APE, EvoPrompt, and APO where applicable) compare if a distinct architecture, such as LLaMA 3.1-8B-Instruct, was used as the Evaluation Model for all methods, similar to the setup used for the portability study?*
>
> Please refer to answer 3) Cross-model generalization in the general answer section, where we show favourable performance of PRL when using Llama3.1-8B-Instruct as generator or evaluator.
>
> Additionally, Table 6 in the main paper already evaluates EvoPrompt (GA and DE) and APE with Llama as the prompt evaluator during test time and Qwen as the prompt generator and evaluator during training time on the text summarization task. We see that PRL and EvoPrompt mainly generalize across models, while APE does less so (compare against Table 2 for the non-cross model generalizability for EvoPrompt and APE).

---

### Official Review · Reviewer_tMpA · 2025-10-27

**Soundness:** 2
**Presentation:** 2
**Contribution:** 2
**Rating:** 2
**Confidence:** 4

**Summary:**

The paper introduces PRL, a RL framework that trains a prompt-generation model to automatically produce effective task prompts. It can be viewed as another R1-style (GRPO RL training) approach applied to the domain of automatic prompt generation.

**Strengths:**

1. Framing prompt optimization as a reinforcement learning problem with a frozen evaluator is clear and straightforward.
2. The method demonstrates consistent empirical improvements across multiple benchmark tasks.

**Weaknesses:**

1. The paper motivates prompt optimization as an underexplored and crucial problem, but recent work (e.g., instruction tuning, preference alignment, RLHF) has substantially reduced the marginal importance of prompt optimization for strong LLMs. The paper would benefit from a deeper discussion or quantitative evidence that prompt optimization still provides meaningful gains for modern aligned models.
2. Most baseline methods cited for automatic prompt generation date back to 2022–2023. It would strengthen the paper to incorporate or discuss more recent developments in this area.
3. The proposed method requires RL training for each dataset, which introduces significant computational overhead compared to evolutionary or heuristic prompt optimizers. The paper lacks a systematic analysis of the resulting latency and compute cost versus accuracy trade-off, making it difficult to assess practical efficiency.
4. The motivation for training a separate prompt generator model is somewhat unconvincing. The paper does not compare against stronger or larger generator models (e.g., Qwen2.5-72B, GPT) or expert-crafted prompts, which could potentially achieve comparable results without RL training.
5. Since the generator must be retrained for every dataset, the method’s scalability and generality are limited. The paper does not explore whether a single generator can generalize across multiple datasets or related domains.
6. The chosen benchmarks (e.g., GSM8K, SAMSum, SST, AGNews) are somewhat outdated and relatively simple. Evaluating on more challenging or recent datasets would better demonstrate PRL’s robustness and contemporary relevance.
7. The ablation on model size lacks a clear causal interpretation. Performance improvements when scaling from 7B to 32B evaluators could largely stem from the inherent capability gain of the larger models rather than from PRL’s prompt optimization.

**Questions:**

Could you include detailed statistics for the training, validation, and test datasets used in your experiments?

---

> ### Author Response · Authors · 2025-11-19
> **Addressing reviewer's concerns (1/4)**
>
> We want to thank the reviewer for the constructive feedback. We address each one of the reviewer's concerns below.
>
> *W1 The paper motivates prompt optimization as an underexplored and crucial problem, but recent work (e.g., instruction tuning, preference alignment, RLHF) has substantially reduced the marginal importance of prompt optimization for strong LLMs. The paper would benefit from a deeper discussion or quantitative evidence that prompt optimization still provides meaningful gains for modern aligned models.*
>
> We thank the reviewer for raising this important point. We agree that instruction tuning, preference alignment, and RLHF reduce prompt brittleness significantly. However, a growing body of very recent work shows that even modern, aligned LLMs still exhibit substantial and often subtle prompt sensitivity, and that explicitly optimizing prompts continues to yield meaningful gains. Below we respond in bullet points and will revise the paper accordingly.
>
> 1. **Recent work shows that aligned LLMs remain highly sensitive to small, semantics-preserving prompt changes.**
>
> * Sclar et al. [1] find that subtle formatting choices (e.g., bullet style, whitespace, placement of labels) can change few-shot accuracy on LLaMA-2-13B by up to 76 accuracy points, and that this sensitivity persists even with larger models.
>
> * Razavi et al. [2] systematically vary only the phrasing of prompts with the same information content and show large, unpredictable accuracy swings across state-of-the-art LLMs.
>
> * Chatterjee et al. [3] and Zhuo et al. [4] show that, even for instruction-tuned chat models, the gap between the best and worst semantically equivalent prompts remains large, and that small rephrasings can significantly alter model behaviour.
>
> * Errica et al. [5] show that, even when average accuracy is high, minor changes in natural-language instructions often flip predictions.
>
> * Cao et al. [6] analyze worst-case prompts and show that for models such as Llama-2-70B-chat the difference between best and worst semantically equivalent prompts can exceed 45 percentage points on RobustAlpacaEval, with worst-case accuracy close to random guessing.
>
> 2. **Small, human-plausible perturbations still hurt robustness.**
>
> * Zhu et al. [7] demonstrate that minor typos, synonym substitutions, and light paraphrases, changes that preserve meaning for humans, can substantially degrade performance of LLMs.
>
> * Li et al. [8] show that short, injected instructions can override original goals in aligned instruction-following models, which implies that specific prompt fragments still exert disproportionate influence despite RLHF.
>
> 3. **Classical in-context learning results already showed the importance of prompt choice.**
>
> * Zhao et al. [9] and Lu et al. [10] show that the choice and ordering of few-shot examples can move GPT-style models from near-chance to near–state-of-the-art performance.
>
> * Min et al. [11] further show that how demonstrations are written often matters more than their exact labels, reinforcing that prompt design itself is a major performance factor.
>
> 4. **Even recent large RL-aligned reasoning models exhibit prompt sensitivity.**
>
> * The DeepSeek-R1 paper [12] explicitly notes that a 671B-parameter RL-trained reasoning model remains sensitive to prompt design, and that seemingly reasonable prompting strategies (e.g., few-shot prompting) can decrease performance.
>
> * The NeurIPS 2025 GRACE framework [13] further underscores this: it uses DeepSeek-R1 as the optimizer LLM and DeepSeek-V3-0324 as the base model for prompt optimization, and shows that purely prompt-level changes yield non-trivial gains over both manual prompts and prior prompt optimizers.
>
> * GRACE explicitly motivates its design by the “high prompt sensitivity” of the base model and uses DeepSeek-R1’s reasoning ability to craft better prompts for DeepSeek-V3.
>
> 5. **Our experiments give additional quantitative evidence on aligned models.**
>
> * We evaluate PRL on instruction-tuned chat models (Qwen2.5-7B-Instruct, Qwen2-14B-Instruct, Qwen2-32B-Instruct, and LLaMA-3.1-8B-Instruct), i.e., models that already underwent instruction tuning / alignment.
>
> * Larger evaluation models (Qwen2-14B and Qwen2-32B) still benefit from PRL over strong manual prompts, indicating that prompt optimization remains useful as model capacity grows.
>
> * Prompts learned on Qwen2.5-7B transfer well to LLaMA-3.1-8B, and retraining PRL directly on LLaMA yields further gains. This suggests that the benefits of prompt optimization are not tied to a single model family.
>
> In summary, the cited literature shows that (i) even instruction-tuned, RLHF’d, and large LLMs remain highly sensitive to subtle, semantics-preserving prompt variations; (ii) this sensitivity affects both average and worst-case performance and is problematic for reliability; and (iii) automatic prompt optimization methods do in fact deliver significant gains on modern aligned models. We will  add this discussion a revised paper.

---

> ### Author Response · Authors · 2025-11-19
> **Addressing reviewer's concerns (2/4)**
>
> *W2 Most baseline methods cited for automatic prompt generation date back to 2022–2023. It would strengthen the paper to incorporate or discuss more recent developments in this area.*
>
> We have evaluated GRACE, a concurrent SOTA automatic prompting work, and present the results in the general section under 2) Comparison with GRACE & hard reasoning problems. We also evaluated PRL on more challenging hard reasoning problems (DeepMath, Math500) and domain-specific problem (MedQA).
>
> *W3 The proposed method requires RL training for each dataset, which introduces significant computational overhead compared to evolutionary or heuristic prompt optimizers. The paper lacks a systematic analysis of the resulting latency and compute cost versus accuracy trade-off, making it difficult to assess practical efficiency.*
>
> Since computational overhead has been raised by multiple reviewers, we refer reviewer to answer 1) Computational cost & generalizability in the general responses section.
>
> The latency during test time is comparable to any other method, since we just prefill the KV-cache with the generated prompt.
>
> Regarding compute cost vs. accuracy trade-off: We agree that our method strikes a different balance for the compute cost vs. accuracy tradeoff than previous work (as regards training time) in that our method may take longer to find a prompt, which is significantly better and more generalizable than those obtained from faster methods. Depending on the application, this might be a favorable trade to make.
>
> *W4 The motivation for training a separate prompt generator model is somewhat unconvincing. The paper does not compare against stronger or larger generator models (e.g., Qwen2.5-72B, GPT) or expert-crafted prompts, which could potentially achieve comparable results without RL training.*
>
> We thank the reviewer for this insightful comment. We agree that a natural question is whether, given a sufficiently strong LLM, one can simply sample many instructions and select a good prompt without any RL training. Our motivation for a separate prompt generator is precisely to systematically search this very large prompt space and compute efficient way, rather than relying on unguided sampling.
>
> To directly address the reviewer’s suggestion, we ran an additional experiment with a much larger generator model. We use Qwen2.5-72B-Instruct only as a prompt generator and keep the evaluation setup identical to the main paper (frozen Qwen2.5-7B-Instruct on SUBJ). We sample 1, 5, 15, 50, and 100 prompts from Qwen2.5-72B-Instruct with the same decoding temperature as in PRL, score them on the training set, select the best prompt, and finally evaluate that prompt on the test set:
>
>
> | # Prompts | Best Prompt ID | Train Acc | Test Acc |
> | --------: | -------------: | --------: | -------: |
> |         1 |              1 |    71.39 |   71.50 |
> |         5 |              2 |    71.74 |   71.65 |
> |        15 |             12 |    74.18 |   74.15 |
> |        50 |             12 |    74.18 |   74.15 |
> |       100 |             12 |    74.18 |   74.15 |
>
>
> PRL average test accuracy on SUBJ: 76.90.
>
> Thus, even when sampling up to 100 prompts from a much larger 72B model, the best selected prompt remains noticeably below PRL, and performance saturates already around 15 samples. This suggests that (i) the prompt space is indeed highly complex, and (ii) guided RL optimization learns to navigate it more effectively than unguided sampling from a very strong generator.
>
> Regarding expert-crafted prompts: in all our experiments we already compare against the MI and NI baselines, which are considered to be good human-crafted prompts. This setup is standard in the automatic prompt engineering benchmarks. PRL consistently and substantially outperforms these human-written prompts (e.g., +19 points on SUBJ), reinforcing the value of training a dedicated prompt generator rather than relying solely on manual design or sampling from a larger LLM.

---

> ### Author Response · Authors · 2025-11-19
> **Addressing reviewer's concerns (3/4)**
>
> *W5 Since the generator must be retrained for every dataset, the method’s scalability and generality are limited. The paper does not explore whether a single generator can generalize across multiple datasets or related domains.*
>
> We thank the reviewer for raising this point. We agree that PRL requires retraining the prompt generator for each task, which does impose a scalability and generality limitation. However, we believe the practical impact of this is mitigated, and we offer the following clarifications:
>
> 1. Task-specific training is a current norm in automated prompt engineering: The majority of prompt optimisation workflows assume a per-task prompt search (i.e., one prompt generator run per dataset) because optimal prompts depend heavily on dataset structure, label space, and domain. Existing work therefore treats “one prompt per dataset” as the standard benchmark for automated prompt engineering.
>
> 2. Once produced, prompts generalise across models, please see our answer 3) Cross-model generalization in the general answers section.
>
> 3. Exploring a single universal generator is out of scope but a promising direction. We acknowledge that we did not investigate whether a single prompt generator can handle multiple datasets or domains simultaneously. While interesting future work, our paper focuses on the widely adopted and fair comparison scenario of “one generator per dataset”.
>
> For other aspects regarding generalizability please refer to 1. Computational cost & task generalizability in general answer section.
>
> *W6 The chosen benchmarks (e.g., GSM8K, SAMSum, SST, AGNews) are somewhat outdated and relatively simple. Evaluating on more challenging or recent datasets would better demonstrate PRL’s robustness and contemporary relevance*
>
> Please refer to 2. Comparison with GRACE & hard reasoning problems in general answer section.
>
> *W7 The ablation on model size lacks a clear causal interpretation. Performance improvements when scaling from 7B to 32B evaluators could largely stem from the inherent capability gain of the larger models rather than from PRL’s prompt optimization.*
>
> We thank the reviewer for this comment. In Table 5 our goal was to demonstrate that prompt engineering remains important even as model size increases. We observe that on the MR dataset the baseline manual prompt accuracy rises from 87.40 → 90.15 when scaling from 7B → 32B parameters (due to larger model capacity). More importantly, PRL also improves consistently: from 91.27 at 7B up to 92.52 at 32B. This shows that, regardless of model size, the model remains susceptible to prompt quality and still benefits from optimized prompts, rather than saturating prompt engineering.
>
> *Q1 Could you include detailed statistics for the training, validation, and test datasets used in your experiments?*
>
> We used exactly the same splits for training/validation and test set across all methods. We have taken the splits from EvoPrompt. They are
>
> | Dataset|Train|Val| Test |
> | -| -| -| -|
> | SST2| 6720   | 200 | 1821 |
> | CR| 1575| 200 | 2000 |
> | MR| 8462| 200 | 2000 |
> | SST5| 8344| 200 | 2210 |
> | AG’s News | 119800 | 200 |7600|
> | TREC| 5252| 200 | 500|
> | SUBJ| 7800| 200 | 2000 |
> | GSM8K| 7273| 200 | 2000|
> | ASSET| 1900| 100 | 359|
> | SAMSum| 718| 100 | 819|

---

> ### Author Response · Authors · 2025-11-19
> **Addressing reviewer's concerns (4/4)**
>
> References
>
> [1] M. Sclar, Y. Choi, Y. Tsvetkov, A. Suhr. Quantifying Language Models’ Sensitivity to Spurious Features in Prompt Design or: How I Learned to Start Worrying About Prompt Formatting. ICLR, 2024.
>
> [2] A. Razavi, M. Soltangheis, N. Arabzadeh, S. Salamat, M. Zihayat, E. Bagheri. Benchmarking Prompt Sensitivity in Large Language Models. ECIR, 2025.
>
> [3] A. Chatterjee, H. S. V. N. S. K. Renduchintala, S. Bhatia, T. Chakraborty. POSIX: A Prompt Sensitivity Index for Large Language Models. Findings of EMNLP, 2024.
>
> [4] J. Zhuo, S. Zhang, X. Fang, H. Duan, D. Lin, K. Chen. ProSA: Assessing and Understanding the Prompt Sensitivity of LLMs. Findings of EMNLP, 2024.
>
> [5] F. Errica, G. Siracusano, D. Sanvito, R. Bifulco. What Did I Do Wrong? Quantifying LLMs’ Sensitivity and Consistency to Prompt Engineering. NAACL, 2025.
>
> [6] B. Cao, D. Cai, Z. Zhang, Y. Zou, W. Lam. On the Worst Prompt Performance of Large Language Models. NeurIPS, 2024.
>
> [7] K. Zhu, J. Wang, J. Zhou, Z. Wang, H. Chen, Y. Wang, L. Yang, W. Ye, Y. Zhang, N. Z. Gong, X. Xie. PromptRobust: Towards Evaluating the Robustness of Large Language Models on Adversarial Prompts. 2024.
>
> [8] Z. Li, B. Peng, P. He, X. Yan. Evaluating the Instruction-Following Robustness of Large Language Models to Prompt Injection. EMNLP, 2024.
>
> [9] T. Z. Zhao, E. Wallace, S. Feng, D. Klein, S. Singh. Calibrate Before Use: Improving Few-Shot Performance of Language Models. ICML, 2021.
>
> [10] Y. Lu, M. Bartolo, A. Moore, S. Riedel, P. Stenetorp. Fantastically Ordered Prompts and Where to Find Them: Overcoming Few-Shot Prompt Order Sensitivity. ACL, 2022.
>
> [11] S. Min, X. Lyu, A. Holtzman, M. Artetxe, M. Lewis, H. Hajishirzi, L. Zettlemoyer. Rethinking the Role of Demonstrations: What Makes In-Context Learning Work? EMNLP, 2022.
>
> [12] D. Guo, D. Yang, H. Zhang, J. Song, R. Zhang, R. Xu, Q. Zhu, S. Ma, P. Wang, X. Bi, et al. DeepSeek-R1: Incentivizing Reasoning Capability in LLMs via Reinforcement Learning. arXiv:2501.12948, 2025.
>
> [13] W. Shi, Y. Chen, S. Bian, X. Zhang, K. Tang, P. Hu, Z. Zhao, W. Lu, X. Du. No Loss, No Gain: Gated Refinement and Adaptive Compression for Prompt Optimization. NeurIPS, 2025.

---

### Official Review · Reviewer_tiGz · 2025-11-01

**Soundness:** 2
**Presentation:** 3
**Contribution:** 2
**Rating:** 4
**Confidence:** 3

**Summary:**

This paper introduces PRL (Prompts from Reinforcement Learning), a reinforcement learning-based
approach to automatically generating and optimizing prompts for large language models (LLMs). PRL uniquely
enables the synthesis of novel few-shot examples not seen during training and integrates explicit reasoning
steps before prompt output. This method achieves excellent empirical performance in tasks such as text
classification. The research scheme includes carefully designed reward shaping, prompt selection, and
detailed ablation experiments.

**Strengths:**

1. PRL devises a clear RL-based prompt optimization loop. Compared to other methods, PRL can create few-
shot prompt examples not limited to the original training data

2. PRL is evaluated across varied tasks, and multiple ablation studies dissect the contribution of prompt
selection, few-shot examples, and explicit reasoning. Additionally, its effectiveness is verified on models of
different architectures and sizes.

**Weaknesses:**

1. Compared to other methods, PRL requires more computational resources for training and has insufficient
generalization ability, which means that PRL needs to be retrained for different tasks, greatly limiting its
usability.

2. There is insufficient discussion on the instability, scalability, and generalization of reinforcement learning:
- Can training on a single task generalize to other tasks? Furthermore, how does the performance and
generalization of simultaneous multi-task learning compare to that of single-task learning?
- Is model training sensitive to the introduced reward function, does reward manipulation exist, and
what is the interaction between format specification rewards and task correctness rewards?
- Is it effective to use a different generator model for training than the one used to evaluation model?
Furthermore, can the same generator model be directly used for different evaluation models after
training?

3. One of the paper's cores is that few-shot prompting behavior "spontaneously emerges" from the RL setup,
yet there is little to no formal analysis or justification. A more rigorous explanation (e.g., does the reward
landscape incentivize synthesis, or is it an artifact of the prompt generator's architecture?) is sorely missing.

4. The compared methods are limited to those before 2023, lacking comparison results with new methods
from 2024-2025.

**Questions:**

see weaknesses.

---

> ### Author Response · Authors · 2025-11-19
> **Addressing reviewer's concerns (1/2)**
>
> We thank the reviewer for the insightful feedback and address the concerns below.
>
> *W1. Compared to other methods, PRL requires more computational resources and must be retrained per task, limiting usability.*
>
> Please see our general response to 1) Computational cost & generalizability.
>
> *W2 There is insufficient discussion on the instability, scalability, and generalization of reinforcement learning*
>
> - Instability: We thank the reviewer for raising these points. We did not observe instability in GRPO training in any of our experiments; optimization was stable across all tasks without signs of divergence or model/reward collapse. We will share optimization logs in a final version of the paper.
>
> - Scalability: PRL scales naturally with the size of the generator model. We tested a smaller generator (Qwen2.5-3B-Instruct) on the SUBJ task and observed the following:
> | Prompt Generator | Accuracy |
> |--------------|----------|
> | Qwen2.5-3B-Instruct            | 74.62 ± 0.83    |
> | Qwen2.5-7B-Instruct            |  76.90 ± 0.95    |
>
>     This indicates that larger generators produce stronger optimized prompts, suggesting that the pipeline scales as expected.
>
> - Generalizablity of RL: The RL training is run with the same hyperparameters across all tasks and there is no task-specific tuning. In this sense, our RL setup generalizes as well.
>
> *W2.1 Can training on a single task generalize to other tasks? Furthermore, how does the performance and generalization of simultaneous multi-task learning compare to that of single-task learning?*
>
> Please also refer to 1) Computational cost & generalizability in the general comment section.
>
> Training on a single task does not in general generalize to other tasks directly, since the produces prompt is by design task-specific, in line with previous work (e.g. APE, EvoPrompt, GRACE etc.), which is the standard evaluation protocol in automated prompt engineering.
>
> Multi-task prompt optimization is an interesting direction but lies outside the scope of this work. We agree that building a general multi-task prompter is an exciting direction for future work.
>
>
> *W2.2 Is model training sensitive to the introduced reward function, does reward manipulation exist, and what is the interaction between format specification rewards and task correctness rewards?*
>
> We appreciate the reviewer’s insightful question. Our reward function consists of two components:
>
> 1. format specification rewards: This encourages the generator to output in the required \<think\>...\</think\> and \<answer\>...\</answer\> structure. This component is identical across all tasks and is not adjusted per dataset.
> 2. task correctness reward: This uses the same metric as the final evaluation (e.g., accuracy or SARI), matching exactly the scoring functions used in APE and EvoPrompt. This ensures fairness and comparability. We do not manipulate or reshape the correctness reward in any way beyond what baselines already employ.
>
> Because the task reward is aligned directly with evaluation, and the format reward is fixed, PRL does not rely on reward shaping that could lead to reward hacking or misalignment.
>
> We note that during RL training both RL reward components go up simultaneously, with format correctness going up already in the beginning, while task correctness grows more slowly.

---

> ### Author Response · Authors · 2025-11-19
> **Addressing reviewer's concerns (2/2)**
>
> *W2.3.1 Is it effective to use a different generator model for training than the one used to evaluation model?*
>
> Yes, please see our answer on 3) Cross-model generalization in the general answers section, specifically point (i).
>
> *W2.3.2 Furthermore, can the same generator model be directly used for different evaluation models after training?*
>
> Yes, please see again our answer on 3) Cross-model generalization in the general answers section, specifically point (ii).
>
> *W3 One of the paper's cores is that few-shot prompting behavior "spontaneously emerges" from the RL setup, yet there is little to no formal analysis or justification. A more rigorous explanation (e.g., does the reward landscape incentivize synthesis, or is it an artifact of the prompt generator's architecture?) is sorely missing.*
>
> We thank the reviewer for this comment. In our setup, few-shot prompting is not explicitly incentivized: we never instruct the generator to produce few-shot examples, and the reward only reflects task performance and formatting constraints. Nevertheless, we consistently observe that the model spontaneously begins to synthesize examples over the course of RL training.
>
> For illustration, below we show the evolution of a sentiment-classification prompt at different training iterations:
>
> | iteration | prompt |
> |----------|--------|
> | 100      | "In this task, you will be given sentences from movie reviews. Your job is to determine whether the sentiment of the sentence is positive or negative. Positive sentiment means the sentence expresses a favorable or good opinion, while negative sentiment means the sentence expresses an unfavorable or bad opinion. Ignore any context outside the sentence itself. Respond only with the label 'positive' or 'negative'." |
> | 500      | In this task, you will classify the sentiment of movie review sentences as either 'positive' or 'negative'. A positive sentiment indicates a favorable or approving view, while a negative sentiment indicates an unfavorable or disapproving view. Carefully analyze the sentence to determine the overall sentiment. Provide only the label ('positive' or 'negative') in your response. |
> | 1200     | In this task, you are given sentences from movie reviews. Your goal is to classify each sentence as 'positive' or 'negative' based on its sentiment. Pay close attention to the context and nuances in the text, as the sentiment might not be explicitly stated. Examples: - "The acting was superb, and the plot was engaging." -> positive - "The movie was so slow and boring that I almost fell asleep." -> negative Return only the label 'positive' or 'negative' without any additional text. |
>
> Early in training (iteration 100/500), the model mainly explores rephrasings of an instruction-style prompt. As training progresses (iteration 1200), it begins to append concrete labeled examples, i.e., few-shot demonstrations, even though these are never mentioned in the instruction or reward. This suggests that, under our reward landscape, the generator discovers that synthesizing examples is an effective strategy to improve task performance.
>
> We view this emergent few-shot synthesis as an important empirical finding and one of the key conceptual contribution of PRL.
>
> *W4 The compared methods are limited to those before 2023, lacking comparison results with new methods from 2024-2025.*
>
> We thank the reviewer for raising this point. In response, we have additionally evaluated our method against GRACE, a concurrent state-of-the-art automatic prompting approach, and our method compares favorably. The new results and discussion are reported in the General Section under “2. Comparison with GRACE & hard reasoning problems".

---

### Author Response · Authors · 2025-11-19
**3. Cross-model generalization**

There are two aspects to cross-model generalization:

(i) Can we train with different combinations of prompt generator and prompt evaluator models and

(ii) after training, can we switch to a different prompt evaluator model without retraining.

PRL is robust at both of these cross-model generalizations. Regarding (i), using different generator and evaluator models can be beneficial. In Table 6 in the paper, we have introduced a PRL-LLaMA variant that uses Llama3.1-8B-Instruct as the generator. The results are reproduced below (the first line corresponds to PRL-Llama in the paper, the second line to PRL):


| Prompt Generator       | Prompt Evaluator (Training) | Prompt Evaluator (Test)      | Rouge-1      | Rouge-2      | Rouge-L      |
|------------------------|-----------------------|-----------------------|--------------|--------------|--------------|
| LLama3.1-8B-Instruct   | Qwen2.5-7B-Instruct         | Qwen2.5-7B-Instruct  | 43.70±0.02   | 16.66±0.25   | 37.46±0.5    |
| Qwen2.5-7B-Instruct    | Qwen2.5-7B-Instruct   | Qwen2.5-7B-Instruct | 42.47±0.83   | 16.17±0.24   | 37.73±0.36   |
| Qwen2.5-7B-Instruct    | Qwen2.5-7B-Instruct | LLama3.1-8B-Instruct | 39.38±2.82 | 15.77±1.56 | 30.57±2.80 |


These results show that PRL is flexible: different generator models can be used without degrading performance, and cross-model training can even yield improvements.

Moreover, for this rebutall we crafted a new experiment in which we train with Qwen2.5-7B-Instruct as prompt generator and Llama3.1-8B-Instruct as evaluator on the summarization task. We present the results in the following table:

| Prompt Generator: Qwen2.5-7B-Instruct | Prompt Evaluator: Llama3.1-8B-Instruct (Train/Test) |
| ------------------------------------------- | --------------------------------------------------------- |

| Method  | ROUGE-1           | ROUGE-2           | ROUGE-L           |
| ------- | ----------------- | ----------------- | ----------------- |
| MI      | 28.92             | 9.29              | 25.09             |
| APE     | 38.47±0.66 | 12.96±0.14        | 33.17±0.22        |
| GA      | 37.48±2.65        | 13.92±1.14 | 33.48±2.54|
| **PRL** | **40.06±0.02**    | **13.98±0.12**    | **34.06±0.01**    |
 |


Regarding (ii), we also see at the table above that prompts produced by PRL using Qwen2.5-7B-Instruct (as both generator and evaluator during training) can be directly applied to Llama3.1-8B-Instruct for evaluation during test time. These transferred prompts achieve state-of-the-art results on the summarization task, demonstrating that PRL produces general prompts that remain effective across evaluation models.

---

### Author Response · Authors · 2025-11-19
**2. Comparison with GRACE & hard reasoning problems**

We want to thank the reviewers for pointing out the need of comparing PRL with more recent baselines and on more challenging benchmarks. Therefore we estimated GRACE [1] (published at NeurIPS2025) with Qwen2.5-7B-Instruct as prompt generator and evaluator.
Following the reviewers' suggestions, we additionally evaluated PRL on three more challenging and recent benchmarks. We chose:

- MATH500 [3] and DeepMath [2] both large-scale, high-difficulty mathematical reasoning benchmarks.
- MedQA [4] is a large-scale multiple-choice medical QA benchmark based on professional board-exam-style questions.

As shown in the tables, PRL consistently outperforms all baseline prompt-optimization methods, including GRACE, on all three benchmarks, indicating that its advantages extend to substantially harder reasoning and domain-specific settings. We would be very happy to evaluate PRL on more additional datasets you would recommend and include those results in the final version. We present the results averaged over 3 runs along with standard deviations.

We performed evaluation on (best scores are in **bold**):

1. Mathematical Reasoning tasks (DeepMath, Math500, GSM8K):

| MATH500| |
| -| - |

|Method| Accuracy|
| -| - |
| APE    | 31.53±1.04 |
| DE     | 34.20±1.39 |
| GA     | 40.13±1.39 |
| GRACE  | 33.20±1.60 |
| PRL    | **44.40±1.40**|

| DeepMath| |
|-|-|

|Method| Accuracy|
| -| - |
|APE| 15.47 ± 0.45 |
|GA| 18.63 ± 2.37 |
|DE| 16.10 ± 0.00 |
| GRACE| 15.05 ± 0.16 |
| PRL | **21.58 ± 0.22** |

| GSM8K | |
| -| - |

|Method| Accuracy|
|-|-|
| MI| 78.20|
| APE | 83.43±1.98|
| GA| 81.62±1.38|
| DE| 79.52±0.45|
| GRACE| 82.37±1.82|
| PRL| **86.15±0.55**|

2. MedQA (domain base task):

|Method |Accuracy|
|-|-|
|APE|45.66 ± 0.97|
|GA|51.95 ± 1.61|
|DE|51.76 ± 0.16|
|GRACE| 52.26 ± 0.16|
|PRL| **53.34 ± 0.11** |


3. Classification tasks:

| Method | CR                  | MR               | SST-5              | AG's News          | SST-2            | TREC               | Subj               | Avg         |
| ------ | - | -| - | -| - | -| --| ----------- |
| MI     | 87.25               | 87.40            | 52.31              | 82.29              | 92.70            | 69.20              | 57.95              | 75.59       |
| NI     | 91.50               | 90.85     | 51.90              | 83.43              | 95.77        | 66.60              | 68.10              | 78.31       |
| APO    | **93.48 ± 0.24**    | 89.97 ± 1.37     | 53.94 ± 0.29       | 83.73 ± 0.31 | 93.71 ± 0.25     | 71.30 ± 1.90       | 69.80 ± 5.96       | 79.42       |
| APE    | 92.87 ± 0.02        | 89.90 ± 0.94     | 49.37 ± 5.66       | 82.58 ± 1.20       | 91.23 ± 0.66     | 77.07 ± 1.61 | 73.92 ± 1.39       | 79.56       |
| GA     | 92.75 ± 0.40        | 90.45 ± 0.72     | 53.76 ± 1.13       | 82.24 ± 1.00       | 94.65 ± 1.04     | **79.20 ± 2.83**   | 74.93 ± 3.12 | 81.14 |
| DE     | 93.38 ± 0.19 | 89.98 ± 0.24     | 55.25 ± 0.37 | 82.18 ± 1.04       | 93.29 ± 0.34     | 76.47 ± 0.38       | 73.08 ± 4.95       | 80.52       |
| GRACE  | 90.92 ± 1.15        | 89.60 ± 1.51     | 53.96 ± 0.93       | 82.34 ± 0.39       | 93.61 ± 0.53     | 72.53 ± 8.62       | 73.92 ± 3.05       | 79.55       |
| PRL    | 92.83 ± 0.24        | **91.27 ± 0.05** | **56.21 ± 0.15**   | **84.36 ± 0.08**   | **96.32 ± 0.04** | 77.07 ± 2.36 | **76.90 ± 0.95**   | **82.14**   |

4. Summarization task:

|Method|ROUGE-1|ROUGE-2|ROUGE-L|
|-|-|-|-|
|MI| 32.76| 10.39| 28.97|
|APE| 37.12±2.02        | 12.97±0.74        | 33.32±1.68       |
|GA| 39.69±1.76| 14.47±1.00        | 35.84±1.63       |
|DE| 33.91±4.04| 12.53±1.47        | 31.05±3.79       |
|GRACE|40.61±0.54| 14.65±0.53        | 35.86±0.54       |
|PRL| **42.47±0.83**| **16.17±0.24**    | **37.73±0.36**   |

5. Simplification task:

|Method|SARI|
|-|-|
|MI| 43.77|
|APE| 45.33±0.83|
|GA| 46.25±0.47|
|DE| 45.79±0.35|
|GRACE| 50.21±0.18|
|PRL| **52.26±3.51**|

We want to note that GRACE was developed with DeepSeek-R1 as prompt generator (called optimizer in their work) and DeepSeek-V3-0324 as evaluator. These are substantially larger and higher performing LLMs than the Qwen-2.5-7B-Instruct models we used. For the results above we have rerun GRACE on Qwen-2.5-7B-Instruct for a fair comparison. We have used the default hyperparameters given by the GRACE authors.


References:

[1] W. Shi, Y. Chen, S. Bian, X. Zhang, K. Tang, P. Hu, Z. Zhao, W. Lu, X. Du. No Loss, No Gain: Gated Refinement and Adaptive Compression for Prompt Optimization. NeurIPS, 2025.

[2] He, Zhiwei, et al. "Deepmath-103k: A large-scale, challenging, decontaminated, and verifiable mathematical dataset for advancing reasoning." arXiv preprint arXiv:2504.11456 (2025).

[3] Lightman, Hunter, et al. "Let's verify step by step." The Twelfth International Conference on Learning Representations. 2023.

[4] Yang, Hang, et al. "Llm-medqa: Enhancing medical question answering through case studies in large language models." arXiv preprint arXiv:2501.05464 (2024).

---

### Author Response · Authors · 2025-11-19
**1. Computational cost & task generalizability**

Below are our answers to issues raised by multiple reviewers:

**Computational Cost:**

We agree that PRL is more compute-intensive than some prior methods. This is a consequence of its deliberately larger search space: whereas APO and EvoPrompt mainly operate by rephrasing a base prompt via critique loops or evolutionary edits (thus constraining exploration), PRL does not impose such restrictions. This broader search is precisely what enables PRL to synthesize novel few-shot examples and incorporate explicit reasoning structures, capabilities that, in turn, drive the observed performance gains.

Crucially, PRL’s compute cost is front-loaded and amortized. Although we report a 48-hour budget, the built-in Prompt Selection mechanism surfaces strong prompts well before the end of training; these intermediate prompts can be deployed immediately. We illustrate this with the evolution of the final prompt’s performance over time:

|CR|
|-|

| Time (hours) | Accuracy |
|--------------|----------|
| 2            | 0.924    |
| 8            | 0.9245   |
| 18           | 0.9255   |
| 24           | 0.930     |
| 48           | 0.930   |

|SST5|
|-|

| Time (hours) | Accuracy |
|--------------|----------|
| 0.9          | 0.516 |
| 1.7          | 0.536 |
| 2.6          | 0.538 |
| 3.4          | 0.544 |
| 6.9          | 0.547 |
| 12.0         | 0.549 |
| 18.0         | 0.554 |
| 23.1         | 0.560 |
| 48.0         | 0.560 |


Even when optimization continues beyond these early checkpoints, the prompts discovered at earlier stages are already competitive and practically useful. Moreover, PRL’s outputs are plain-text prompts, so there is no inference-time overhead relative to manually engineered prompts.

The general trend on other datasets is similar.

We want to note that GRACE, another concurrent SOTA prompter, for which we also provide additional experiments below, is also training for up to 48 GPU hours for some of the tasks that require longer output, e.g. summarization, simplification and DeepMath.

**Task generalization:**

In line with the accepted evaluation protocol and similar to existing work our prompts are task-specific and do not generalize across tasks. Since PRL also synthesizes few-shot examples, which drives improvement in task performance, this task-specificity cannot be easily avoided.

In summary, PRL does use more compute because it searches a richer space that enables novel few-shot synthesis and explicit reasoning. However, Prompt Selection provides early, deployable prompts; inference remains as cheap as any manual prompt; and the learned prompts transfer across evaluators (Table 6). We believe this yields a favorable trade-off between computational cost and practical usability.

---

### Author Response · Authors · 2025-12-02
**Summary Comment**

We thank the Area Chair and reviewers for their time and constructive feedback. The comments were invaluable in clarifying and strengthening our work. Below we summarize the strengths and concerns raised by the reviewers, and how we addressed them during rebuttal.

**I. Reviewers highlighted several key strengths of PRL:**

1. Strong empirical performance and generalization, including SOTA results across diverse tasks (classification, summarization, simplification, math reasoning) (tiGz, tMpA, vKbp).

2. Synthesis of new few-shot examples not present in the training data (tiGz, vKbp).

3. Comprehensive ablations and analysis (tiGz, vKbp).

4. Clarity and writing quality (vKbp, rKJG).

5. Robustness across model architectures and sizes (tiGz, vKbp).

6. Explicit reasoning integration (tiGz, vKbp).

7. Clear RL-based optimization loop (tiGz, tMpA, vKbp).


**II. Summary of major concerns raised by reviewers:**
1. Computation cost (tiGz, tMpA, vKbp, rKJG):
Additional experiments show that prompts from early training stages are already competitive. While PRL uses more compute, this reflects its richer search space that enables novel few-shot synthesis and explicit reasoning.

2. Comparison with most recent methods (tiGz, tMpA):
We added comparisons with a recent SOTA-baseline GRACE (Neurips 2025). PRL outperforms it in all cases when using the same LLM. Additionally, we conducted evaluations on math (Math500 and DeepMath) and medical reasoning (MedQA), where PRL sets a new SOTA controlling for LLM.

3. Cross-model generalization (tiGz, vKbp, rKJG):
Experiments with Llama as generator/evaluator show that PRL is flexible: swapping generator models does not harm performance and cross-model training can even improve it.

4. Instability, scalability, and generalization of reinforcement learning (tiGz, tMpA):
Conclusions from our experiments:
    - no instability in GRPO training.
    - the larger generators produce stronger optimized prompts, suggesting that the pipeline scales as expected.
    - the RL training is run with the same hyperparameters across all tasks and there is no task-specific tuning. In this sense, our RL setup generalizes as well.

5. Sensitivity to reward function (tiGz):
    As the reward is aligned directly with evaluation, and the format reward is fixed, there is no possibility for reward hacking or misalignment.

6. Few-shot examples analysis (tiGz, vKbp):
    We present an experiment showing that PRL does not generate examples immediately at the start of the training, but synthesizes them later on. This suggests that the generator discovers that synthesizing examples is an effective strategy to improve task performance.

7. Importance of prompt optimization (tMpA):
    We cite recent literature (13 articles) that provide evidence that even modern, aligned LLMs still exhibit substantial and often subtle prompt sensitivity, and that explicitly optimizing prompts continues to yield meaningful gains.


**III. Summary of minor concerns raised by reviewers:**

1. The motivation for training a separate prompt generator model (tMpA):
    Sampling prompts from a larger, frozen Qwen model produced worse prompts than PRL, showing the necessity of training the generator.

2. Ablation on model size lacks a clear causal interpretation (tMpA):
    Our experiments on model size show that, regardless of model size, the model remains susceptible to prompt quality and still benefits from optimized prompts, rather than saturating prompt engineering.

3. Dataset statistics (tMpA):
    We provided detailed statistics for the training, validation, and test datasets. These splits were used across all methods.

4. Path to a Universal Prompt Generator (tMpA, vKbp):
    We provided two possible future directions towards PRL as a Universal Prompt Generator: Meta-learning / fast adaptation and pretraining on large data corpora.

5. Ablation on the influence of the “thinking process” (rKJG):
    We described our experiment showing that the removal of the reasoning phase leads to a significant drop in accuracy.

6. Base prompt concerns (rKJG):
    i) Sensitivity: Using an optimized PRL prompt as the base prompt yields similar performance, showing low sensitivity.
    ii) Iterative refinement: Periodically replacing the base prompt gave no clear gain.
    iii) Generating prompts from scratch: Some minimal task specification is required and our current manual base prompts meet this condition; points (i) and (ii) show that the base prompt is not a bottleneck.

---

### Meta-Review · Area_Chair_8FMh · 2026-01-06

**Summary:**

This paper introduce PRL, which trains a prompt-generator model with reinforcement learning (GRPO-style) to automatically produce effective prompts for a frozen evaluator LLM, sometimes synthesizing new few-shot examples and adding explicit reasoning structure.

The reviewers acknowledge strong empirical performance and clear implementation, but raise recurring concerns about conceptual novelty, practicality, and scalability. Several reviewers view the method as a straightforward application of existing RL (GRPO-style) optimization to prompt search, with limited new algorithmic insight. Compute cost and the need to retrain the prompt generator per task are seen as major limitations, especially compared to heuristic or evolutionary prompt optimizers. There is also concern that the main distinctive claim, emergent few-shot synthesis, is not sufficiently explained or analyzed beyond empirical observation. These issues, taken together, weigh against acceptance despite solid results.

**Reviewer Concerns:**

Addressed or partially addressed by the rebuttal:

- Recency of baselines and benchmark coverage: The authors added comparisons to a recent baseline (GRACE) and evaluated on harder reasoning and domain benchmarks, which meaningfully strengthens the empirical positioning.

- Cross-model generalization: Additional experiments show that prompts can transfer across evaluator models and that different generator–evaluator combinations work reasonably well.

- Motivation for a trained generator vs. sampling: The added experiment with a much larger generator model provides evidence that unguided sampling saturates below PRL.

- Ablations on reasoning and components: Clarifications and ablations help establish that the reasoning phase and prompt selection contribute materially.

Outstanding concerns:

- Conceptual novelty: The core approach remains close to existing RL-based optimization pipelines, and the paper does not clearly articulate a new algorithmic principle beyond system-level orchestration. The emergence of few-shot synthesis, while interesting, lacks a convincing mechanistic explanation.

- Practical efficiency and scalability: The method remains computationally expensive and requires per-task retraining. The rebuttal improves the narrative but does not provide a decisive cost–benefit justification relative to simpler methods.

- Generality across tasks: Prompts are task-specific and the generator does not amortize across datasets; cross-model transfer does not resolve this core limitation.

- Evaluation framing: Even with added experiments, the gains are demonstrated mainly on settings where prompt sensitivity is already known, leaving uncertainty about broader impact.

**Reviewer Scores:**

Reviewer tiGz (4): Likely unchanged at 4. While many technical concerns were addressed, the novelty and scalability issues remain.

Reviewer tMpA (2): Likely unchanged at 2. The rebuttal addresses missing comparisons and benchmarks, but does not resolve the reviewer’s fundamental skepticism about novelty, practicality, and the importance of prompt optimization.

Reviewer vKbp (6): Likely unchanged at 6. Positive on empirical strength and few-shot synthesis, but still aware of compute and generality limitations.

Reviewer rKJG (6): Likely unchanged at 6. Main requests (ablations and cross-model tests) were addressed, but not enough to justify a higher score.

---

### Decision · Program_Chairs · 2026-01-26

Reject